# Replicability in Earth System Models

Kai R. Keller[1], Marta Alerany Solé[1], and Mario Acosta[1]

[1]Barcelona Supercomputing Center, Barcelona, Spain

**Correspondence:** Kai R. Keller (kai.keller@bsc.es)

**Abstract.** Climate simulations with Earth System Models (ESMs) constitute the basis of our knowledge about the projected climate change for the coming decades. They represent the major source of knowledge for the Intergovernmental Panel on Climate Change (IPCC), and are an indispensable tool in confronting climate change. Yet, ESMs are imperfect and the climate system that they simulate is highly non-linear. Therefore, small errors in the numerical representation, initial states, and boundary conditions provided for future scenarios, quickly develop large uncertainties in the trajectories of their projected climates. To improve the confidence and minimize the uncertainty, future projections use large ensembles of simulations with the same model, and ensembles with multiple models. Using these two types of ensembles concurrently addresses two kinds of uncertainty in the simulations, (1) the limited spatiotemporal accuracy of the initial states, and (2) the uncertainty in the numerical representation of the climate system (model error). The uncertainty for the future development of the anthropogenic climate drivers is addressed by projecting different Shared Socioeconomic Pathway (SSP) scenarios. Organizing multimodel ensembles to make confident statements about the future climate addressing different SSP scenarios is a tremendous collaborative effort. The Coupled Model Intercomparison Project (CMIP) addresses this challenge, with the participation of 33 modeling groups in 16 countries. As one among numerous challenges that such undertaking poses, we are addressing model replicability in this article. The anticipated number of simulated years in the 6th CMIP phase (CMIP6) accumulated to about 40,000 years. With typical values for the computational throughput of about 1 to 15 simulated years per day (SYPD), it is clear that the simulations needed to be distributed among different clusters to be completed within a reasonable amount of time. Model replicability addresses the question, whether the climate signal from different scientific scenarios generated by the same model, performed on different clusters, can be attributed exclusively to the differences in the scientific drivers. It has been shown, that even changing specific compiler flags, leads to significant changes in certain climatological fields. Model replicability holds, when the model climatologies derived from the same model under a different computing environment, are statistically indistinguishable. If replicability does not hold, we cannot be certain that differences in the model climate are exclusively attributed to differences in the scientific setups. In this article, we present a novel methodology to test replicability. We further establish an objective measure of what constitutes a different climate based on Cohen's effect size. We provide a thorough analysis of the performance of our methodology and show that we can improve the performance of a recent state-of-the-art method by 60%. We further provide an estimate of the ensemble size that is required to prove replicability with confidence. We find that an effect size of $d = 0.2$ can be used as a threshold for statistical indistinguishability. Our analysis, based on the Community Earth System Model 2 (CESM2) Large Ensemble Community Project (LENS2) 100-member ensemble, shows that with 50 members, we can resolve effect sizes of $\sim 0.3$, and with ensembles of 20 members, we can still resolve effect sizes of $\sim 0.35$.

We further provide a robust methodology to objectively determine the required ensemble size, depending on the purpose and
requirement of the replicability test.

## 1    Introduction

Since the last three assessment reports issued by the Intergovernmental Panel on Climate Change (IPCC), and the Paris Agree-
ment in 2015, besides the scientific community, the global community, stakeholders, and policymakers seem to have broadly
accepted the reality of climate change. In Europe and the United Kingdom, the funding for climate science and specifically
climate change related projects has increased significantly since the 4th IPCC assessment report (AR4) in 2007 from about 10
to almost 140 million US\$ in 2020 (Sovacool et al., 2022; AbdulRafiu et al., 2022). The 6th IPCC assessment report (AR6)
projects that 1 in 50 years heat waves, become about 8 times more frequent, and 1 in 10 years extreme precipitation events,
become 1.5 times as frequent in a 1.5-degree warmer climate compared to the pre-industrial period (Pörtner et al., 2022).
The future climate will undergo significant regional changes, and we will experience an increased intensity in certain natu-
ral disasters such as floodings, storms, and droughts (Theochari and Baltas, 2024; Krichen et al., 2024; Edamo et al., 2023).
The scientific community is given considerable resources by public entities to develop climate mitigation strategies, for in-
stance, through transformations within the energy sector, and to develop solutions for adapting to climate change (Sovacool
et al., 2022). The committed anthropogenic climate impact is already noticeably changing global weather patterns, which
is why adaptation strategies become increasingly important. Traditional policies for agricultural or other land usage do not
apply anymore, posing a challenge for the global food security, especially in regions that already suffer food shortage and
hunger (Wheeler and Von Braun, 2013). Besides developing strategies for climate mitigation and adaptation, it is important to
continuously monitor and reassess the impact of anthropogenic activity on the future climate with updated scenarios.

The most prominent initiative dedicated to this aim is the Coupled Model Intercomparison Project (CMIP) (O'Neill et al.,
2016), which constitutes one of the largest global initiatives dedicated to gaining a deeper understanding of climate change
across past, present, and future periods within a multi-model framework. This is a highly complex and ambitious endeavor
involving weather and climate centers from around the world. In the most recent phase (CMIP6), a total of 21 model inter-
comparison projects (MIPs) were endorsed, encompassing 190 distinct experiments that entail 40,000 years of simulation and
approximately 40 PB of storage, which represents a large cluster of data (Acosta et al., 2024; Eyring et al., 2016). This exten-
sive range of experiments allows for a comprehensive analysis of climate change projections by comparing results from various
models. However, organizing such a large-scale initiative is a monumental task, and the lack of proper traceability in exper-
iments could severely impact the consistency of model-to-model and configuration-to-configuration comparisons (including
different configurations within the same model). If key factors such as reproducibility and replicability do not hold, conclu-
sions based on experiments with the same model conducted among different research centers are disputable. While possibly
afflicted with uncertainties caused by running on different computing platforms, or using a different level of computational
optimizations, differences cannot be exclusively attributed to the differences between the scientific scenarios.

Reproducibility, in here, achieving identical results when conducting the same experiment under the same conditions by two different experimenters, is important in this collaborative effort, tracing back simulations (specific configurations, model versions, compilation flags, etc.) to reproduce the same simulation in the same environment, achieving identical results. Reproducibility in climate science can be achieved using automatic workflow managers such as Autosubmit (Manubens-Gil et al., 2016; Uruchi et al., 2021) or ecFlow (Bahra, 2011), and the findability, accessibility, interoperability, and reusability (FAIR) standard has been developed to achieve the reproducibility of scientific results (Wilkinson et al., 2016). Equally important is replicability, achieving identical results when performing the same experiment configuration using different environments. Replicability is much more challenging, and in practice, bit-to-bit replicability is in the best case hard or computationally expensive, but often impossible to achieve (Geyer et al., 2021). However, results can be replicable in the sense that the model's climate in one computing environment is statistically indistinguishable compared to results from simulations performed in another environment.

It has been demonstrated that the non-linearity of the models leads to significantly different trajectories, even for a different set of compiler flags (Döscher et al., 2022). Massonnet et al. developed a state-of-the-art replicability test that was successfully applied to different EC-Earth configurations and platforms within the CMIP6 context (Massonnet et al., 2020; Döscher et al., 2022). However, the increasing complexity of ESMs community models (such as EC-Earth) with multiple coupled components and higher resolutions, as well as the new requirements for consistency and robustness in the upcoming initiatives or projects such as CMIP7, Destination Earth (Hoffmann et al., 2023), and Earth Virtualization Engines (EVE) (Stevens et al., 2024), have created the need to reassess available replicability methods and improve their accuracy and insight, to ensure the replicability of all experiments across different computing environments and platforms with confidence.

Based on robust and well-established statistical methods, we established a methodology to evaluate the performance of the replicability test from Massonnet et al., and developed an improved methodology including three new metrics and three new statistical tests. We will demonstrate that all the metrics and tests are robust against false rejection, and, based on this, present a new methodology combining all the methods and metrics into a single replicability test. We show that with this new methodology, we can improve the accuracy towards Massonnet's test by about 60%. For instance, when testing replicability with 5-member ensembles, our methodology is capable of resolving differences of 1 standard deviation in contrast to 1.7 standard deviations with Massonnet's methodology. We further use the Community Earth System Model 2 (CESM2) Large Ensemble Community Project (LENS2) climate ensemble to establish a threshold for when statistical differences are significant and climate model results non-replicable. Our analysis with two 50-member ensembles drawn from the full 100-member LENS2 ensemble (Rodgers et al., 2021) shows that ensembles of that size, while stemming from the same configuration, yet show an effective difference of up to 0.16 standard deviations. With this, we are able to objectively assess the performance of our methodology. Besides, employing the LENS2 ensemble, we demonstrate that with a 20-member ensemble, we can resolve statistical differences of about 0.36, and with a 50-member ensemble differences of at least 0.3 standard deviations. Additionally, based on Cohen's effect size (Cohen, 1988b), we establish an objective measure of the statistical differences between two model climates.

The remainder of the article is as follows: In Section 2, we address the research question and detail on the necessity of testing for replicability, and discuss other work and the state-of-the-art. In Section 3, we explain our methodology for the replicability test, and for the assessment of its performance. We further explain the statistical concepts and introduce Cohen's effect size that builds the foundation for assessing what constitutes a different climate. In Section 4, we present our findings, and conclude the article in Section 5

## 2 Motivation and state of the art

A consequence of the non-linearity of climate and weather models is their sensitivity to small variations of the initial state (Mihailović et al., 2014; Lorenz, 1962, 1963). The transformation of the model equations into the binary number system, leads to quantization errors that propagate and grow along the model's trajectory. Moreover, floating point operations are not commutative, and do not comply to the associative property. Consequently, changing the order of the operations changes the result.
Therefore, using a different compiler on the same architecture, or different levels of optimizations with the same compiler, will slightly change the outcome of the simulation (Geyer et al., 2021; Massonnet et al., 2020; Reinecke et al., 2022). Any change within the computing environments affects the simulated trajectory at every time step, introducing small errors which propagate and grow, and eventually lead to an entirely different outcome. Nonetheless, model comparison across different systems and modeling teams requires that the models generate the same climate among different environments. For instance,
when comparing future projections following different emission scenarios, it is essential, that we can attribute the differences in the climatological fields exclusively to the different set of forcings and boundary conditions. If replicability does not hold, signals from the forcings and from the change in computing environment will mix, and we cannot be certain of the conclusions we draw from such experiments. This necessity has been recognized in a number of research groups, and a number of studies exist on the topic (Baker et al., 2014, 2015, 2016; Mahajan et al., 2017; Milroy et al., 2018; Mahajan et al., 2019; Massonnet
et al., 2020; Zeman and Schär, 2022; Reinecke et al., 2022; Price-Broncucia et al., 2025).

To decide whether a climate in one environment is the same as in another environment, we employ the concept of statistically significant difference. To account for the model uncertainties, we run not one, but an ensemble of simulations. Ensembles are generated by providing a sample of perturbed initial states $\Phi_0^{(m)} = \Phi_0 + \delta_m$, with $m = 1, \ldots, M$, and $M$ the ensemble size. The perturbation, $\delta_m$, is typically applied to the temperature fields of either the ocean or the atmosphere. What we require
from a simulation performed within a different environment is that the ensemble distribution in environment A is statistically indistinguishable to the one in environment B. If this holds, we say the climate of the model in environment A is replicable in environment B.

Most of the existing methodologies addressing replicability consider simulations of one year or less. Keeping the simulations short, is motivated by the cost that long ensemble simulations cause. In addition, the tests that are described in the literature are
125 almost exclusively designed for single component models, i.e., forced atmosphere, land, or ocean components. For instance, Baker et al. present an ensemble consistency test for the atmospheric component (CAM-ECT) of the CESM (Baker et al., 2015). The test is based on a 151-member ensemble of 1-year simulations generated using the established model version, and

an additional 3 ensemble members with a model version subject to validation. The methodology uses principal component analysis (PCA) to provide linearly independent variables that form the basis for the statistical analysis. Baker et al. observe that the test correctly captures statistical consistency where it is expected, and also where it is expected to fail. In a later article, Milroy et al. revises the same methodology and proves that a 350-member ensemble of very short simulations, integrating the model for only 9 time steps, is sufficient for the methodology to work accurately. Similar tests exist for ocean components. However, replicability of ocean components cannot be tested with the same methodology, as the processes in the ocean are much slower than in the atmosphere. The available methodologies agree that a minimum of 1 year is required to verify statistical consistency between two different model versions (Baker et al., 2016; Mahajan et al., 2017).

In the context of continuous integration and model development, replicability methodologies for single components are useful. Stringent requirements on low computational cost and short time-to-completion prohibit long climate simulations with large ensembles for each new model update. However, a distinction needs to be made between acceptable model climates and comparable model climates. During model development, we need to relax the requirements on the methodologies to test for acceptable model climates. However, when comparing the climates of different model versions to identify effects of different climate forcings, it is essential that the climates are identical under the same scientific constraints. Otherwise, our conclusions are biased with artificial signals. For instance, short simulations in controlled climate scenarios are inappropriate to evaluate climate drivers with interannual variability. This is acknowledged by Baker et al., suggesting that tests such as the POP-ECT are not suitable to evaluate the statistical consistency of the ENSO interannual variability (Baker et al., 2016). Short tests with isolated ESM components cannot address slowly evolving processes and dynamic feedbacks arising in coupled models. Evaluating statistical consistency of long climate simulations with fully coupled models require different test designs. This is the motivation behind methodologies such as the one we are presenting here and the methodology from Massonnet et al.

Model replicability is a very active field of research. Yet, until to date, it is unclear if the existing methodologies can be extended to resolutions higher than the typical 100 Km grid spacing. Methodologies such as the CESM ensemble consistency test (CESM-ECT) (Baker et al., 2015) and the work from Zeman (Zeman and Schär, 2022) require ensemble sizes to be in the order of 30 to 150 members. Although, in Baker's test the ensemble for the trusted ensemble is created only once for each new version of the code, and the test for replicability only requires 3 ensemble members for the validation in a different environment, it becomes prohibitive at very high resolution to even generate a 150-member ensemble only once. While the Ultra-Fast CAM Ensemble Consistency Test (UF-CAM-ECT) (Milroy et al., 2018) makes operational testing for high-resolution atmospheric components viable, it is not applicable to the much slower ocean component and coupled models. The same applies for the methodology presented by Zeman et al.. Both Baker and Zeman acknowledge that until today it is not clear if short simulations should be used for testing coupled models subject to dynamic feedbacks and slowly evolving ocean processes. While the cost of replicability methodologies for coupled models, entailing the question of the required simulation length, is important, especially for application to high-resolution models, and certainly needs to be addressed in the future, it is not within the scope of this article.

On the positive side, climate simulations at Km-scale become more common with the increasing computing power of our supercomputers. Initiatives such as DYnamics of the Atmospheric general circulation Modeled On Non-hydrostatic Domains

(DYAMOND) (Stevens et al., 2019) and Destination Earth (Hoffmann et al., 2023; Rackow et al., 2025) extend the limits for high-resolution climate simulations at 5 to 10 Km horizontal resolution, making it conceivable to perform long ensemble integrations in the near future. Further, climate models, at 25 to 100 Km resolution, are still the standard and extensively used to date. The upcoming 7th CMIP version (CMIP7) still considers simulations at 25 Km resolution for the High Resolution Model Intercomparison Project 2 (HighResMIP2) (Roberts et al., 2024) and 100 Km resolution for most of the other MIPs. Consequently, ensemble-based methodologies for long climate integrations remain important, while it is necessary to develop those methodologies further.

The methodology from Massonnet et al. targets the replicability of fully coupled ESMs conducting multi-decadal climate simulations and builds the foundation of our contribution. The method is based on the calculation of a single score for the climatology of each member of an ensemble of simulations. Two ensembles are created, one in environment A and another in environment B. This leads to two samples of scores, one sample for each ensemble. The authors then apply a two-sample statistical test with the null hypothesis, that both samples are drawn from the same probability distribution. The authors can detect differences with statistical significance, only if the sample means are apart by more than 2 standard deviations. Our analysis shows that this is not enough to reliably detect statistical indistinguishability. We employ Cohen's effect size, an objective statistical measure for statistical difference of two samples, to estimate, when two ensembles of simulations are statistically indistinguishable. We show that the effect size between two ensembles generated with the same model, within the same computing environment, and with the same configuration and boundary conditions, hence with identical climates, lies between 0.11 and 0.16. Based on this, we developed a new methodology that extends the one from Massonnet et al. by new metrics and statistical tests, increasing the accuracy of the test while reducing the required ensemble size. Note that the range for the effect size given above was established using the CESM and the LENS2 ensemble. It is unclear if using a different model would lead to different effect sizes when comparing identical climates.

## 3 Methodology

The structure of the replicability test that we propose (see Figure 1) follows an approach based on Massonnet et al. (2020). Initially, we generate two ensembles of $M$ climate simulations with a specific ESM in two different computing environments. By computing environment, we refer to a certain compiler version or vendor, computer architecture, or underlying software library versions such as the Message Passing Interface (MPI) or math libraries such as LINPACK and BLAS. The test also considers different model versions, compiler options, software updates, etc. The test targets any kind of technical change that is not intended to change the model climate, but could potentially do so. By ensemble, we mean a set of simulations conducted by the same climate model within the same computing environment, where each simulation only differs by small, climate-invariant perturbations in the initial conditions. The aim is to describe the climate system using a representative distribution, with an ensemble large enough to capture the key features of the underlying population of possible model states, and simulations long enough to capture the internal climate variability. Once the ensembles are created, we compute a score $s_m^X$ for each ensemble member $m$ and variable $X$, yielding a distribution of scores $\{s_m^X\}_{m=1,\ldots,M}$, for each output variable and ensemble. We then

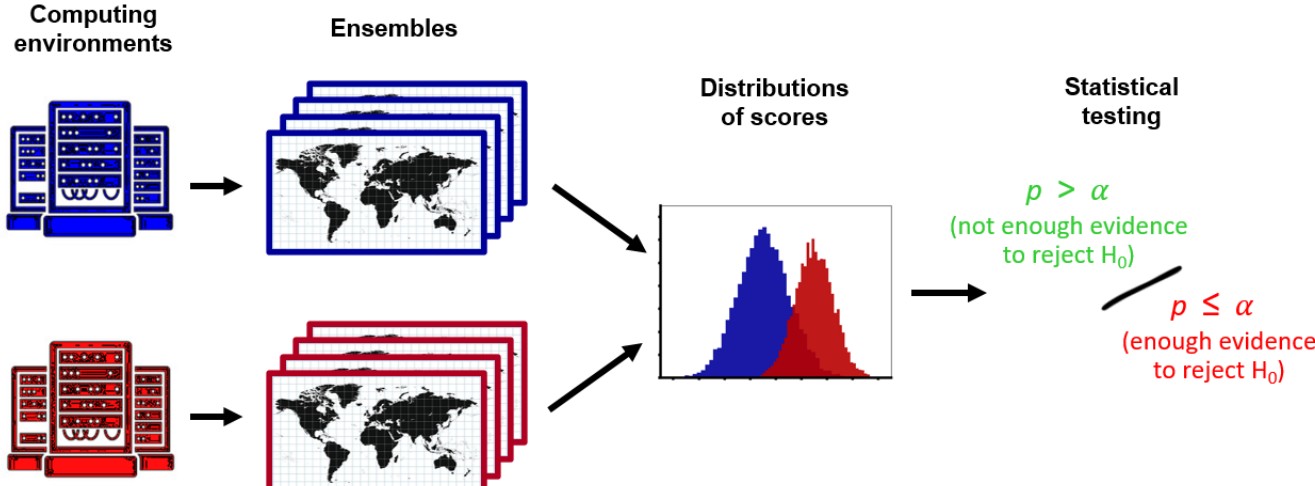

**Figure 1.** Schematic of the replicability test structure: we port a given ESM to two different computing environments; by executing the model with a fixed forcing and introducing small, climate-invariant perturbations to the initial conditions, we generate one ensemble of simulations for each environment; we apply a metric to the resulting ensembles, generating one score per each ensemble member; this results in two distributions of scores that we compare employing statistical tests that determine whether there is enough evidence to reject the null hypothesis $H_0$ at a certain significance level $\alpha$.

apply a statistical test to the two distributions of scores for each variable, with the null hypothesis that the two ensembles are statistically indistinguishable, i.e., representing the same climate. Based on the $p$-value returned from the test and a significance level $\alpha$, we can make an informed decision to either reject or not reject the null hypothesis $H_0$.

The single value score, $s_m^X$, is derived as follows: we first compute the climatology $\bar{X}_m$, i.e., the temporal average for variable
$X$ over the relevant period of time, yielding $N$ values $(\bar{X}_m)_i := \bar{x}_{mi}$ for each member, $m$, with $i = 1, \ldots, N$, and $N$ being the number of grid points. In representing the climatological fields by a single score, we aim to reduce the degrees of freedom, while retaining as much of the spatial information as possible. If we were to directly spatially average the data, important information related to regional differences between the members might be lost. A simple example is two global fields with the same mean, but regional differences that are opposite in sign, whose contribution is therefore averaged out. To avoid this, we
apply a suitable metric to the grid point components of the climatology $\bar{X}_m$, obtaining a relative rather than absolute value. By relating the spatial components of the variables to a reference common for all members, for instance, the mean of the entire simulated ensemble at each grid point, or an observational climatology, we can encode regional differences within the global average of the metric. For instance, a cold temperature is not simply leveled out by a warm temperature, if the bias for the warm region is small, and the bias for the cold region is large. We evaluate various metrics that can potentially be used for
this step, which we will introduce in the next section. The metrics yield a transformed spatial field, from which we calculate the area-weighted average to get the member's score $s_m^X$ for variable $X$. In contrast to the methodology in (Massonnet et al.,

2020), we employ several scores, and not just a single one. Moreover, the scores that we employ are valued between 0 and 1, hence, compatible in their value range. This allows us to link the scores into a combined score, $\tilde{s}$, by:

$$\tilde{s} = \frac{1}{S} \sum_i^S s_i \tag{1}$$

Finally, we apply all the tests to all the available scores, and reject the null hypothesis, if the minimum p-value:

$$\tilde{p}\text{-value} = \min_{\forall s,t \in S,T} p\text{-value}(s,t) \tag{2}$$

is smaller than the significance level $\alpha$. Here, $S$ and $T$ denote the sets of the scores and tests respectively.

### 3.1 Scores

In the following, we will introduce the four scores that our replicability methodology is based on. We try to keep the meaning
of the score intuitive, such that values close to 0 correspond to a bad representation of the model compared to observations and values close to 1 a good representation. This is motivated by the International Land Model Benchmarking (ILAMB) System (Collier et al., 2018), introducing scores specific to assess land model scientific performance. Two scores from ILAMB are of particular interest for our work: the bias-score and the root-mean-square (RMS) error (RMSE) score. We will introduce both of the scores in this section. We further employ an adaptation of the extensively used Reichler-Kim index (Reichler and
Kim, 2008), and the RMS Z-score (RMSZ).

### 3.1.1 Root-mean-square Z-score

The first metric that we use is based on the standard score, or Z-score, motivated by Baker et al. (2014, 2016). Applied to our case, the Z-score measures the deviation of the climatology, $\bar{X}_m$, of member $m$ to the ensemble mean in units of the ensemble standard deviation. Therefore, the Z-score of variable $X$ at grid point $i$ for ensemble member $m$ is given by:

$$Z_{mi}^X = \frac{\bar{x}_{mi} - \mu_i}{\sigma_i}, \tag{3}$$

where $\bar{x}_{mi}$ is the $i$-th component of climatology $\bar{X}_m = \frac{1}{T} \sum_{t=T_0}^{T} X_{mt}$ with $T$ being the length, $T_0$ the starting point of the time period for the climatology, and $X_{mt}$ the variable field at time $t$. The ensemble mean, $\mu_i$, and standard deviation, $\sigma$ at grid point $i$, are given by:

$$\mu_i = \frac{1}{M} \sum_{m=1}^{M} \bar{x}_{mi}, \quad \sigma_i = \sqrt{\frac{1}{M} \sum_{m=1}^{M} (\bar{x}_{mi} - \mu_i)^2}. \tag{4}$$

Accounting for different areas at the grid points, we compute the area-weighted **root mean square Z-score (RMSZ)** for variable $X$ and member $m$ as:

$$\text{RMSZ}_m^X = \sqrt{\sum_{i=1}^{N} \widetilde{\omega}_i \cdot \left(Z_{mi}^X\right)^2}, \tag{5}$$

where $\widetilde{\omega}_i = \omega_i / \sum_{i=1}^{N} \omega_i$ are the normalized area weights with $\omega_i = \cos(\text{lat}_i)$ defined by the latitude at the corresponding cell. To make this metric more robust to small values of the standard deviation $\sigma_i$, and to constrain it within the range $[0,1]$, we compute the final **RMSZ-score (eRMSZ)** as:

$$e\text{RMSZ}_m^X = \sqrt{\sum_{i=1}^{N} \widetilde{\omega}_i \cdot \exp\left[-\left(Z_{mi}^X\right)^2\right]}. \tag{6}$$

### 3.2 The adapted Reichler-Kim index score

Note that the RMSZ-score relies only on the shape of the considered ensemble's distribution and does not require any external reference, making it easy to compute. Nonetheless, if we had two ensembles with a similar spread but centered around different means, the two distributions of the Z-scores would be alike, and a statistically significant difference very difficult to detect. To address this issue, we further explore the performance indices proposed by (Reichler and Kim, 2008) and later on applied in (Massonnet et al., 2020). In contrast to the Z-score, where we relate the grid point components to a quantity derived from the respective ensemble itself, hence, different for each of the two ensembles, for the Reichler-Kim index, we relate the components to an external field, which is the same for both ensembles. This external field is given by an observational dataset, or more specifically, the mean and standard deviation of an observational climatology, $(\bar{Y})_i := \bar{y}_i$, computed from $Y = \{y_{ti}\}_{t=1,\ldots,T,\,i=1,\ldots,N}$. Given the simulated climatology $\bar{x}_{mi}$ for variable $X$ and ensemble member $m$, we define the **adapted RK-index (RK08)** as the area-weighted average of the normalized error variance:

$$\text{RK08}_m^X = \sqrt{\sum_{i=1}^{N} \widetilde{\omega}_i \cdot \left(\frac{\bar{x}_{mi} - \bar{y}_i}{\sigma_i^y}\right)^2}, \tag{7}$$

where the mean $\bar{y}_i$ and standard deviation $\sigma_i^y$ of the observations are calculated over time as follows:

$$\bar{y}_i = \frac{1}{T}\sum_{t=1}^{T} y_{ti}, \quad \sigma_i^y = \sqrt{\frac{1}{T}\sum_{t=1}^{T}(y_{ti} - \bar{y}_i)^2}. \tag{8}$$

Note that the original definition of the Reichler-Kim index requires a reference dataset, for instance, a different ESM simulating the same period using the same boundary conditions. We do not require this for the replicability test, as it would be just a constant multiplier, and would leave the distribution of the scores invariant. As before, the **RK08-score (eRK08)** is computed by applying the exponential to the metric at the grid point values, to make it more robust against small values of the standard deviation $\sigma_i^y$, and constrain it to values within the interval $[0,1]$:

$$e\text{RK08}_m^X = \sqrt{\sum_{i=1}^{N} \widetilde{\omega}_i \cdot \exp\left[-\left(\frac{\bar{x}_{mi} - \bar{y}_i}{\sigma_i^y}\right)^2\right]}. \tag{9}$$

#### 3.2.1 Bias and root-mean-square error scores

In addition, we analyze two more metrics similar to the RK-index and adapted from (Collier et al., 2018), namely, the **bias score (eBIAS)** and the **root-mean-square error score (eRMSE)**. In climate science, the bias is defined as the deviation of

the simulated from the observed climatology from a certain period. With the simulated climatology $\bar{x}_{mi}$ for variable $X$ and ensemble member $m$, and the observed climatology $\bar{y}_i$, the bias is given by:

$$\text{bias}_{mi}^X = \bar{x}_{mi} - \bar{y}_i. \tag{10}$$

Collier et al. divide the bias by the observation's standard deviation, relating the absolute bias to the climate variability or observation error. Hence, the contributions of large biases in regions with a strong variability are reduced, avoiding artificial deterioration of the score. Adopting the definition from (Collier et al., 2018), we compute the exponential of the relative bias and take the area-weighted average to compute the bias score as:

$$\text{eBIAS}_m^X = \sum_{i=1}^N \widetilde{\omega}_i \cdot e^{-|\text{bias}_{mi}^X|/\sigma_i^y}. \tag{11}$$

The RMSE score is defined similarly, but using the centralized root-mean-square error ($\text{crmse}_{mi}^X$) instead of the bias. Following (Collier et al., 2018), we compute the RMSE score as:

$$\text{eRMSE}_m^X = \sum_{i=1}^N \widetilde{\omega}_i \cdot e^{-\text{crmse}_{mi}^X/\sigma_i^y}, \tag{12}$$

where

$$\text{crmse}_{mi}^X = \sqrt{\frac{1}{T} \sum_{t=1}^T [(x_{mit} - \bar{x}_{mi}) - (y_{ti} - \bar{y}_i)]^2}. \tag{13}$$

There are only subtle differences in the three latter scores. The eBIAS score and eRK08 score are fairly similar. The only difference is, that in case of the eBIAS score, the regional differences to observations enter linearly, while in case of the eRK08 score they enter quadratically. This has the consequence, that differences are a bit more pronounced in case of the eRK08 score. While eRK08 and eBIAS scores only consider the differences in the climatological mean state, the eRMSE includes differences to the observational dataset resolved in time. Therefore, we expect the eRMSE score to be more sensitive to temporal differences within the trajectory of the climate simulation.

### 3.3   FPR, power, and effect size

Our assessment of the quality of the replicability test is driven by three quantities: **false positive rate (FPR)**, **true positive rate (power)**, and **effect size (d)**. A robust statistical test is characterized by a small FPR and a high power. It is common to require the FPR to be 5% and the power 80%. The FPR is calculated as FP/(FP+TN), where FP is the number of false positives and TN is the number of true negatives. Hence, it measures the probability that the test will falsely reject the null hypothesis when two samples come from the same distribution. In contrast to that, the power is calculated as TP/(TP+FN), where TP is the number of true positives and FN is the number of false negatives, and represents the probability of correctly rejecting the null hypothesis when the populations are different.

### 3.3.1 Power Analysis and Effect Size

To give an objective measure of the differences between two samples, we employ Cohen's effect size, $d$, a statistical quantity relating the differences in the sample means to a standard deviation:

$$d = \frac{\mu_1 - \mu_2}{\sigma}, \tag{14}$$

where $d$ is the effect size and $\sigma$ the pooled standard deviation defined by (Cohen, 1988a):

$$\sigma = \sqrt{\frac{(n_1 - 1)\sigma_1^2 + (n_2 - 1)\sigma_2^2}{n_1 + n_2 - 2}} \tag{15}$$

$$= \sqrt{\frac{\sigma_1^2 + \sigma_2^2}{2}} \quad , \quad \text{for } n_1 = n_2. \tag{16}$$

With $n_1$ and $n_2$ representing the respective sizes of sample 1 and sample 2. The effect size $d$ quantifies the mean-difference between two samples in units of the standard deviation. Cohen categorized the differences between the samples as small for $d \leq 0.2$, i.e., when the difference of the sample means is smaller than $0.2$ standard deviations. But this is not a general statement, and should be assessed on a case-to-case basis. Nevertheless, we will see later that the effect size can be used as an objective measure for the statistical indistinguishability of two samples. Yet, statistically significant differences between two samples can only be objectively verified with sufficiently large ensemble sizes. This has been described in (Cohen, 1988a) introducing the concept of power analysis. Given the effect size $d$ between two statistically different samples, a power analysis provides the minimum sample size that is required to correctly reject the null hypothesis within a desired confidence interval. The power for a significance level of 5% is typically required to be at least 80%, meaning, we wrongly not reject the null hypothesis in less than 20% of the cases. There exist formulas for determining the sample size for normally distributed random variables. We will show that this formula can be used to some extent as well for complex models. We will demonstrate this based on a chaotic toy-model in Section 4.1.2, and later by performing a power analysis employing Monte Carlo methods for real climate data in Section 4.2.1.

### 3.4 Statistical tests

As previously mentioned, the last step of the test consists in comparing the distributions of scores among the two ensembles. The goal is to determine whether they are statistically indistinguishable, i.e., whether the members of both ensembles originate from the same distribution. Since the true distribution is unknown, and we cannot generally assume that the scores follow a normal distribution, we rely on nonparametric tests, as they do not make any assumptions about the distribution of the data. In particular, we apply four different statistical tests, whose null hypotheses are briefly outlined below.

### 3.4.1 Common statistical tests

We employ 3 well-established statistical tests for our methodology. Firstly, we consider the two-sample **Kolmogorov-Smirnov test (KS-test)**. In this test, the null hypothesis is precisely that the two given samples are drawn from the same distribution.

Thus, it is suitable for evaluating a distribution's shape or spread. Secondly, we consider **Welch's -test (T-test)**, a variant of Student's -test, a two-sample location test with the null hypothesis that the means of the two given samples represent the same population mean. While the Student's -test assumes both samples follow a normal distribution and the variances are equal, it has been shown to be quite robust against violations of the normality assumption (Posten, 1984). However, Welch's -test is the more appropriate choice, as it does not make assumptions about the kind of distribution. Lastly, we consider the **Mann-Whitney -test (U-test)** with the null hypothesis that given randomly selected values $x$ from one and $y$ from the other sample, the probability of $x$ being greater than $y$ is equal to the probability of $y$ being greater than $x$. This means that, similar to the T-test, the U-test is a location test, but it tests for the median being the same, rather than the mean. The choice of these tests was motivated by (Zeman and Schär, 2022; Massonnet et al., 2020), where one can find a more detailed description of the tests.

### 3.4.2 Bootstrap test

Besides the common statistical tests from above, we designed a fourth statistical test for our methodology, based on Efron's bootstrap method (Efron, 1992). The bootstrap method is mainly used to estimate the mean and variance of a population. The method works as follows: given a sample $S$ with sample size $s$, we resample $m$ times from $S$ to generate new samples $S'_m$ drawn from $S$ with replacement. The sample size $s'$ of $S'_m$ remains the same, hence, $s' = s$. As we draw from $S$ with replacement, some of the constituents of the new sample $S'_m$ will be duplicated. For each sample $S'_m$, we compute the mean and store it. This leads to the bootstrap sample $B$ containing the $m$ means computed during the resampling steps. The values within the limits of the 2.5% and 97.5% quantiles of the distribution provide a 95% confidence interval for estimating the true population mean. Similarly, we can estimate the standard deviation by the bootstrap sample standard deviation. The method can also be used to estimate other quantities from an underlying distribution. We use the method to generate a distribution of the effect size $d$ between two sets of scores. We will properly introduce the effect size in Section 3.3.1. We will define the $p$-value of the **bootstrap test (B-test)** based on the null hypothesis that the scores are drawn from the same population:

$$H_0 : \lim_{s \to \infty} \bar{d} = 0. \tag{17}$$

Which corresponds to the statement that the effective differences of the two samples converge to 0 for very large sample sizes. According to the central limit theorem, the distribution of sample means is normally distributed, independently of the distribution of the random variable. We base our premise for the test on this statement, and assume that, if the scores in the joint set are drawn from the same population, the effect size of two sets of scores is normally distributed around 0. Therefore, the null distribution is given by:

$$P(d)\Big|_{H_0 = \text{true}} = \mathcal{N}(0, \sigma_p), \tag{18}$$

where $d$ is the effect size, and $\sigma_p$ the standard deviation of the distribution of effect sizes for two subsamples drawn from the population. We estimate the distribution of effect sizes between random draws of ensembles from the population as $\sigma_B$, i.e., the standard deviation of the bootstrap distribution. Considering the effect size as the random variable resulting from random

draws of two ensembles from the same population, we can define the $p$-value for the null hypothesis as:

$$p\text{-value} = \begin{cases} 2\,P(d \leq \bar{d}), & \text{if } \bar{d} \leq 0 \\ 2\,P(d \geq \bar{d}), & \text{if } \bar{d} > 0 \end{cases}, \tag{19}$$

where $\bar{d}$ is the effect size computed from the bootstrap sample. In other words, assuming the scores from the joint set are drawn from the same population, hence, the effect size is normally distributed around 0, the $p$-value is the probability of finding an effect size $d \geq |\bar{d}|$. After defining the significance level $\alpha$, we can reject $H_0$, if this probability is smaller than $\alpha$.

## 4 Evaluation Section

In the following section, we present the assessment and validation of our replicability methodology. Besides the analysis itself, we will also address the question of what it means that two climates are statistically different. We start by exploring the concepts using two simple models. The first is just a Gaussian sample. The second uses a set of 1000-member ensembles that we generate with the Lorenz-96 model, which will be introduced in detail later. Each ensemble with the Lorenz-96 model is generated with a different forcing, which allows us to assess the performance of the replicability test for different effect sizes using very large ensembles. Following the analysis with the simple models, we exemplify our replicability methodology using real climate data that we select from the LENS2 ensemble, which we will properly introduce later. The first example compares 50 members of the LENS2 ensemble to other 50 members with a slightly different radiative forcing applied. The second example compares 20 members of the LENS2 ensemble to other 20 members using different initial conditions.

### 4.1 Evaluating the Metrics and Tests Based on Toy Models

With the following study based on data drawn from a Gaussian distribution and data from a chaotic toy-model, we (a) show that the relationship between power and sample size holds for both normally distributed data, and data stemming from more complex models, and (b) we calculate the power and FPR of our statistical tests (see Section 3.4) for normally distributed data and data from the chaotic toy-model.

#### 4.1.1 Toy model one: Normal distributions

In this first straightforward evaluation, we estimate the FPR using normally distributed data. In particular, for each execution of the test, we draw two ensembles of the corresponding size from the same normal distribution (with $\mu = 0$ and $\sigma = 1$). We repeat the test 1000 times for various ensemble sizes with a fixed significance level of $\alpha = 5\%$. All four statistical tests exhibit similar FPR values around the significance level, with no clear dependence on the ensemble size (see Figure 2). The FPR of all the tests individually is of about 5% or less. However, combining all four tests yields larger values, of up to 9% when combining all four tests.

For computing the power, we consider ensembles drawn from two normal distributions with the same standard deviation ($\sigma = 1$) but different means differing by up to 4 standard deviations ($\mu_1 = 0.1, 0.2, \ldots, 4\sigma$ and $\mu_2 = 0$). Thus, the effect size in

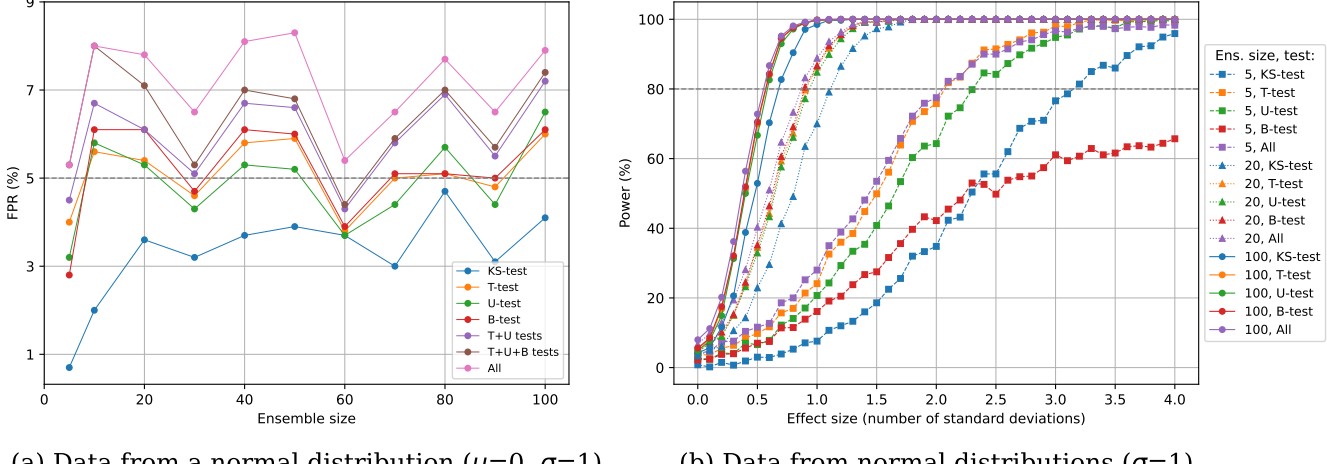

(a) Data from a normal distribution ($\mu$=0, $\sigma$=1)    (b) Data from normal distributions ($\sigma$=1)

**Figure 2.** (a) FPR in terms of the ensemble size for multiple statistical tests and combinations of them. (b) Power in terms of the effect size for various statistical tests and ensemble sizes (squares correspond to 5 ensemble members, triangles to 20, and circles to 100). All data points are estimated by applying the replicability test with $\alpha = 5\%$ to ensembles drawn from normal distributions. Note that 'All' corresponds to combining all four statistical tests with and 'or' statement.

all cases is simply the mean of the first distribution: $d = (\mu_1 - 0)/1 = \mu_1$. With these, we follow the same procedure as for the FPR, running each test 1000 times for each pair of different ensembles and for various ensemble sizes. Figure 2 displays the resulting power. For very small ensembles (5 members), we can only reject $H_0$ with confidence, i.e., reach a power of 80%, if the data shows an effect size of $d \approx 2$ for the T and U-tests and $d \approx 3$ for the KS and B-tests. For the largest ensembles

(100 members), we achieve a power of 80% already for effect sizes of $d \leq 0.5$. This shows that 5 ensemble members, as used in previous studies (Massonnet et al., 2020), might not be a statistically significant sample to obtain reliable results. An important observation is that the performance of the different tests is similar for large ensembles, while the highest power always corresponds to that of the T-test. Further, combining the T-test with other tests does not noticeably increase the power (compare purple with orange data points), but increases the FPR. This suggests that a selection of fewer tests, such as the T

and the U-test, might be the best choice for testing replicability between two samples.

### 4.1.2 Toy model two: Lorenz-96

Next, we conduct analyses upon large ensembles generated with the Lorenz-96 model, a simple toy model designed to investigate error propagation and predictability in climate models (Lorenz, 1996). Particularly, it is a one-dimensional atmospheric model that describes a single scalar quantity, $X = \{x_1, \ldots, x_N\}$, as it evolves on a circular array of $N$ cells, equivalent to a

circle of fixed latitude. Its equations, one for each of the $N$ cells, are given by

$$\dot{x}_i = (x_{i+1} - x_{i-2})x_{i-1} - x_i + F, \tag{20}$$

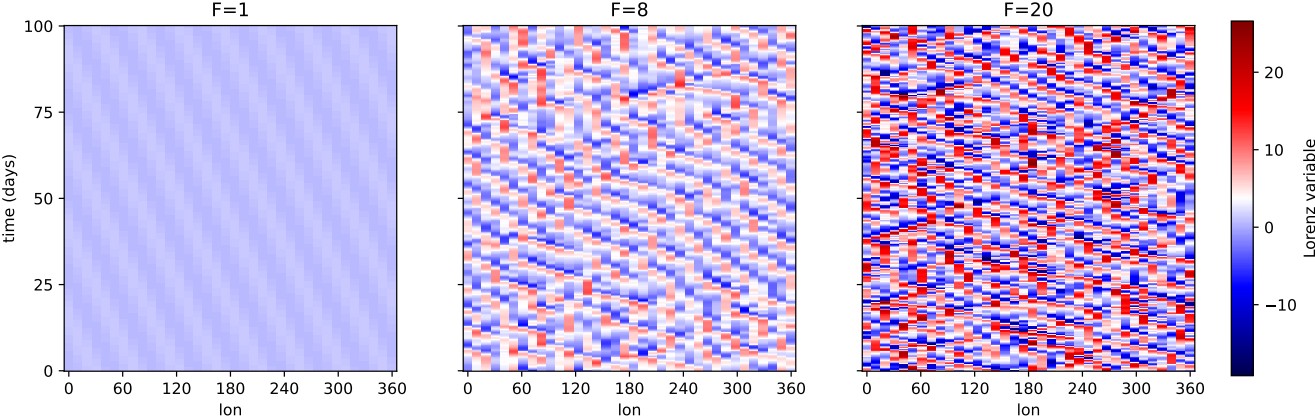

**Figure 3.** Evolution of a circular array of $N = 36$ cells following the Lorenz-96 model for increasing values of the forcing $F$ illustrating different regimes. Larger values of $F$ lead to larger values of the variables and a more chaotic regime.

with the additional condition $x_{i-N} = x_{i+N} = x_i$ for $i = 1, \ldots, N$, such that the model has circular symmetry. The differential equations in Equation 20 are composed of a constant term simulating an external forcing, $F$, a linear term related to internal dissipation, and quadratic terms simulating advection. The evolution of all components $x_i$, is formally the same, without geographical variations (spatial homogeneity), hence, all variables expose statistically similar behavior. The character of the evolution of $X$ in time, is determined by the forcing parameter $F$. Small values of $F$ lead to periodically traveling wave solutions, while larger values lead to a chaotic behavior (see Figure 3).

As noted earlier, conducting tests with this model allows us to generate very large ensembles with many more members than usually feasible with complex climate models within an appropriate limit of computing hours. This allows us to properly analyze the impact of the ensemble size on the replicability study. Moreover, with Lorenz-96 we can easily create ensembles representing different climates by tuning the forcing parameter $F$, allowing for a preliminary assessment of the sensitivity of the tests by estimating their power.

We generated two types of data series with the Lorenz-96 model. The first type serves as surrogate for observations, while the other represents the climate simulations. In all cases, we use 36 longitudes, i.e., $N = 36$, so that each cell covers $10°$ of longitudes, and the observational data series is based on the forcing $F = 8$, so we operate in the chaotic regime. These values are common choices for applications of the model similar to ours (van Kekem, 2018). We span a period of 10 years, while each time unit in Lorenz-96 ($\Delta t = 1$) equals to 5 days (Lorenz, 1996). We use a time-step of $\Delta t = 0.025$ (or 3 hours) for the observations and $\Delta t = \frac{0.025}{3}$ (1 hour) for the simulations. The ensembles are generated by introducing a small perturbation to the initial state, i.e., we add a Gaussian random error of $\mathcal{O}(10^{-3})$ to one of the longitude cells.

We generated 21 1000-member ensembles with different values for the forcing parameter $F$. Conversely, for the observations, we only run the model once with the forcing parameter set to $F = 8$. In either case, the values of $X$ are averaged monthly before saving the data. The ensembles are generated varying the value of the forcing around $F = 8$, both in the periodic and

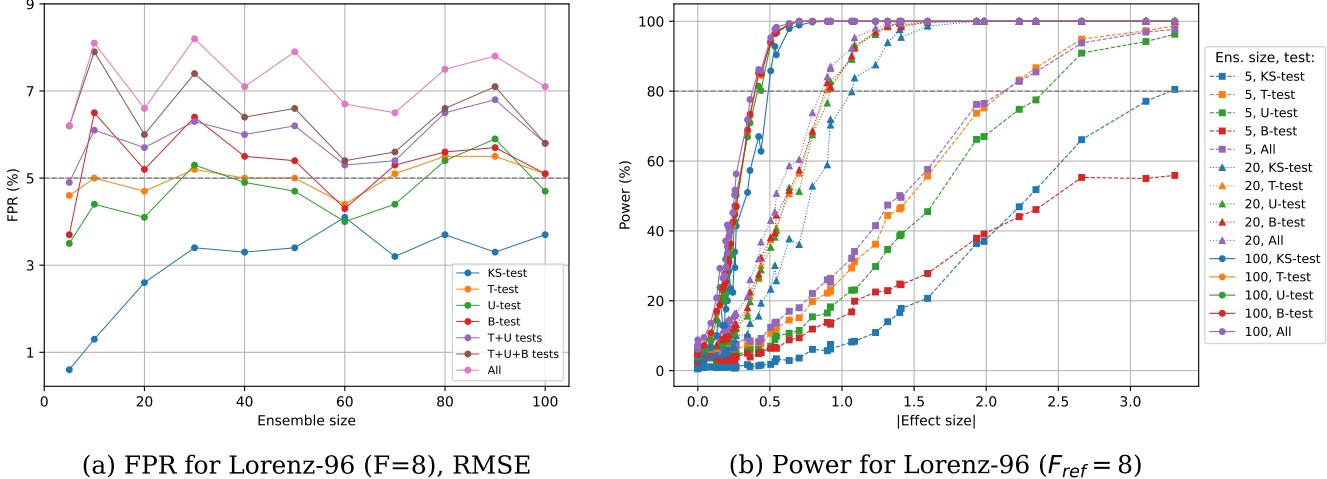

**Figure 4.** (a) FPR in terms of the ensemble size for multiple statistical tests and combinations of them. (b) Power in terms of the absolute effect size for various statistical tests and ensemble sizes (squares correspond to 5 ensemble members, triangles to 20, and circles to 100). All data points are estimated by applying the replicability test with $\alpha = 5\%$ to ensembles drawn from the large 1000-member Lorenz-96 ensembles with different forcings, comparing distributions of scores computed applying the RMSE metric and all metrics, respectively. Note that 'All' corresponds to combining all four statistical tests with an 'or' statement.

chaotic regimes: $F=\{1, 5, 6, 7, 7.5, 7.6, 7.7, 7.8, 7.9, 8.0, 8.1, 8.2, 8.3, 8.4, 8.5, 9, 10, 20, 30, 40, 50\}$. We observed that the model needs some time to stabilize, this is why we performed a short spin-up of 6 months before starting the actual simu-
lations. Executing the model while varying the forcing shows that the values of $F$ influence the ensemble mean (higher for larger forcing). However, varying the forcing around $F = 8$, in the range $F = 7$ to $F = 9$, does not significantly impact the overall trend over time, i.e. the relative evolution of the climate is the same in all cases. By modifying the forcing within this range, we can adjust the overlap between different ensembles, which translates into controlling the effect size with respect to the reference ensemble with $F = 8$ and, hence, allows us to test the power of the replicability test for a scope of effect sizes
similar to the one used in Figure 2 for the normal distributions. Assuming that the 1000-member ensembles generated with the respecting forcings are sufficiently large to reflect the real distribution of climatological states, we can assign an effect size to the comparison between two ensembles with different forcings. The effect sizes, that we compute, correspond to the difference in the distributions of scores among the two ensembles, rather than the values at the longitudes.

Comparing Figure 4 with Figure 2 shows that the results for the FPR and the power in the much more complex case of
the Lorenz-96 model are very close to the FPR and power from the normal distributed samples. This indicates that the scores that we compute for the ensembles are normally distributed for the Lorenz-96 model. Furthermore, in both cases, the power depends strongly on the effect size between two samples. Therefore, we can conclude that if we know the effect size between two ensembles, we can compute the sample size that is required to detect significant differences between the samples with confidence, using an ordinary power analysis, even for the more complex ensembles stemming from the Lorenz-96 model. We

**Table 1.** Climate variables analyzed from the LENS2 dataset with the corresponding sources of observational data used to evaluate the metrics for each considered period. Note that the net_sfc variable is computed as the sum of three quantities: the net surface sensible and latent heat fluxes, and the net surface radiation flux.

| Variable | Description | Units | Observations period 1850-1880 | Observations period 1990-2014 |
|---|---|---|---|---|
| tas | 2m air temperature | K | NOAA 20th Century Reanalysis (V3) | ERA5 (Hersbach et al., 2023) |
| psl | Sea level pressure | Pa | NOAA 20th Century Reanalysis (V3) | ERA5 (Hersbach et al., 2023) |
| prlr | Total precipitation rate | m/s | NOAA 20th Century Reanalysis V3 | ERA5 (Hersbach et al., 2023) |
| ta | Air temperature | K | NOAA 20th Century Reanalysis (V3) | ERA5 (Hersbach et al., 2023) |
| ua | Zonal wind | m/s | NOAA 20th Century Reanalysis (V3) | ERA5 (Hersbach et al., 2023) |
| va | Meridional wind | m/s | NOAA 20th Century Reanalysis (V3) | ERA5 (Hersbach et al., 2023) |
| tauu | Zonal surface stress | Pa | - | ERA5 (Hersbach et al., 2023) |
| tauv | Meridional surface stress | Pa | - | ERA5 (Hersbach et al., 2023) |
| hus | Specific humidity | kg/kg | NOAA 20th Century Reanalysis (V3) | ERA5 (Hersbach et al., 2023) |
| net_sfc | Net surface heat flux | $W/m^2$ | NOAA 20th Century Reanalysis (V3) | ERA5 (Hersbach et al., 2023) |
| tos | Sea surface temperature | K | - | ERA5 (Hersbach et al., 2023) |
| sos | Sea surface salinity | g/kg | - | UKMO EN4.2.2 (Good et al., 2013) [1] |
| siconc | Sea ice concentration | $0 - 1$ (fraction) | NOAA 20th Century Reanalysis (V3) | NOAA 20th Century Reanalysis (V3) |

(1) EN.4.2.2 data were obtained from https://www.metoffice.gov.uk/hadobs/en4/ and are ©British Crown Copyright, Met Office, 2013, provided under a Non-Commercial Government Licence http://www.nationalarchives.gov.uk/doc/non-commercial-government-licence/version/2/.

will see later that the power between Lorenz-96 and the much more complex LENS2 data are again very similar, which lets us extend this statement also to ensembles of complex climate model simulations.

## 4.2    Evaluating the Metrics and Tests Based on the LENS2 Ensemble

The Community Earth System Model 2 (CESM2) Large Ensemble Community Project (LENS2) provides open access to multi-decadal climate simulation data at 1-degree horizontal resolution, conducted with a large ensemble comprising 100
members (Rodgers et al., 2021). The simulation covers a historical period (1850-2014) and a future projection (2015-2100), following the Shared Socioeconomic Pathways (SSP) scenario SSP3-7.0. Important in the context of this study is as well to mention, that the entire ensemble has been performed on the same cluster, i.e., the Aleph Supercomputer at the IBS Center for Climate Physics (ICCP) (Rodgers et al., 2021). Therefore, we can be certain, that differences in the climates are exclusively attributed to differences in the respective members scientific configuration. The variables that we used for our study, and the
respective observational datasets, are listed in Table 1. The LENS2 ensemble is interesting for our analysis because of the composition of the members. The first and second half of the ensemble, 50 members each, use a slightly different biomass burning (BMB) emission forcing. Other than that, the two sub-ensembles use an almost identical setup. While the first 50 members use the original CMIP6 prescribed BMB forcing, the second half uses a smoothed version. The forcing only differs

**Table 2.** LENS2 ensemble composition based on member initialization and biomass burning (BMB) emissions forcing.

| Members | BMB forcing | Initialized in year(s) from pre-industrial control | Initial state uncertainty based on |
|---|---|---|---|
| 1-10 | C6BMB | 1001, 1021, 1041, 1061, 1081, 1101, 1121, 1141, 1161, 1181 | Different years from control |
| 11-20 | C6BMB | 1231 (strong AMOC / decreasing) | Perturb potential temperature (atm) |
| 21-30 | C6BMB | 1251 (average AMOC / decreasing) | Perturb potential temperature (atm) |
| 31-40 | C6BMB | 1281 (weak AMOC / increasing) | Perturb potential temperature (atm) |
| 41-50 | C6BMB | 1301 (average AMOC / increasing) | Perturb potential temperature (atm) |
| 51-60 | smC6BMB | 1231 (strong AMOC / decreasing) | Perturb potential temperature (atm) |
| 61-70 | smC6BMB | 1251 (average AMOC / decreasing) | Perturb potential temperature (atm) |
| 71-80 | smC6BMB | 1281 (weak AMOC / increasing) | Perturb potential temperature (atm) |
| 81-90 | smC6BMB | 1301 (average AMOC / increasing) | Perturb potential temperature (atm) |
| 91-100 | smC6BMB | 1011, 1031, 1051, 1071, 1091, 1111, 1131, 1151, 1171, 1191 | Different years from control |

during the period from 1997 to 2014 and is identical during the remaining period. This represents an interesting use case for our replicability methodology, as we can use it to test its sensitivity. Additionally, the ensemble members use slightly different methods for the initialization. All members are initialized from a certain year of a 1400-year pre-industrial control (PI-control) simulation. However, 20 members (1-10 and 51-60) start from 20 different years of the PI-control (macro perturbation), while the remaining four sets of 20 members start from four different years of the PI-control and are perturbed by adding small random variations to the atmospheric potential temperature field at initialization (micro perturbation) (Rodgers et al., 2021). Those differences are also listed in Table 2. Two subsets of the ensemble that are perturbed differently are particularly interesting for our evaluation. 20 members from the ensemble are initialized from year 1231 of the PI-control, characterized by a strong current associated with the Atlantic meridional overturning circulation (AMOC). Another 20 members are initialized from year 1281 from the PI-control, characterized by a weak current associated with the AMOC. We apply the replicability test on these two samples to test for significant differences between the climates initialized from the two contrasting AMOC phases.

### 4.2.1 Quantifying statistical differences with Cohen's effect size and the LENS2 ensemble

To make statements about the performance of our tests and scores, we need an objective measure for quantifying the differences between two datasets. Based on the 100-member LENS2 ensemble, we will show that Cohen's effect size can serve as such an objective measure. Cohen and others gave an empiric scale for categorizing the differences between two datasets based on the effect size $d$ (Equation 14): *very small* ($|d| \leq 0.01$), *small* ($0.01 < |d| \leq 0.2$), *medium* ($0.2 < |d| \leq 0.5$), *large* ($0.5 < |d| \leq 0.8$), *very large* ($0.8 < |d| \leq 1.2$), and *huge* ($1.2 < |d| \leq 2.0$) (Cohen, 1988b; Sawilowsky, 2009). Can we apply this scale to climate model simulations as well? To answer this question, we compute the effect size between two ensembles generated during a control and a reference period. We compute ensemble-mean climatologies for members 1-50, forced with the original CMIP6 BMB emission forcing, hereinafter C6BMB, and for members 51-100, forced with the smoothed CMIP6

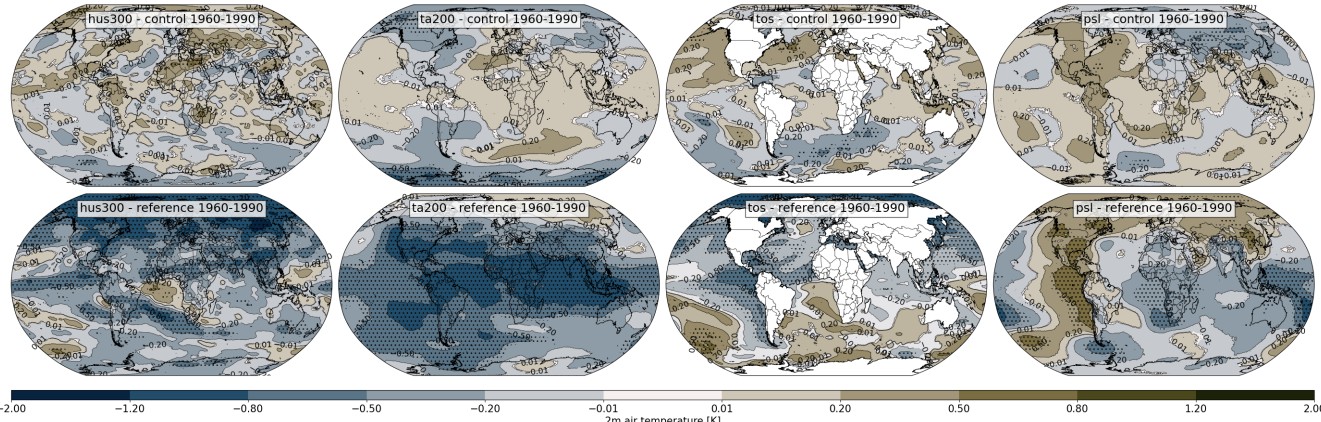

**Figure 5.** Spatial plots of the effect size between the two LENS2 sub ensemble, comparing the default CMIP6 BMB forcing to the smoothed forcing. We show plots for the variables hus300, ta200, tos, and psl, for the control period 1960-1989, where the forcing is the same (top), and for the reference period 1990-2014, where the forcing is different (bottom).

BMB emission forcing, hereinafter smC6BMB. For both ensembles, we compute climatologies from the control period 1960-1989, and from the reference period 1990-2014. As the forcing is only different during 1997 to 2014, and after more than 100 years of simulation, effects from the different member initialization dissipated, we expect the two 50 member sub-ensembles from the control period to be statistically indistinguishable. The sub-ensembles from the reference period, on the contrary, should express significant differences (Rodgers et al., 2021; Fasullo et al., 2022; Heyblom et al.). In Figure 5, we show spatial plots of the effect size between the two ensemble-mean climatologies for a few variables from the reference period (top) and control period (bottom). For those plots, the effect size is calculated based on the actual value of the climatological fields at the grid points, and not from the score or metrics. Interestingly, the majority of the values for $d$ in the control period, are of a magnitude smaller than 0.2. We observe a few values larger 0.2, and also larger than 0.5, but those are very rare. In the plots from the reference period, on the other hand, we observe a significant amount of effect sizes with a magnitude larger than 0.5. To better visualize the distribution of effect sizes on the global domain, we show the cumulative density function for both cases in Figure 6. Our first observation is that for the control, the median of the effect size is for all variables between 0.11 and 0.16, indicated by the shaded area in green. For the reference, the median varies between 0.15 and 0.38. Further, for all variables of the control, less than 5% of the effect size is larger than 0.2 and almost zero values are larger than 0.5. For the reference period, on the other hand, a fair amount of the variables show about 50% of the effect sizes with values larger than 0.2, and up to 20% with values of more than 0.5. Just based on this plot, we can already see that some variables from the reference period show significant differences between members forced with C6BMB and smC6BMB. The internal variability of the climate variables makes it difficult to draw a hard line for statistical indistinguishability, yet, as the median effect size between the control ensembles is in the order of 0.11-0.16, it is reasonable to assume that two climatological datasets that show a median effect size of above 0.2, are significantly different. As a consequence, the metrics and tests that are

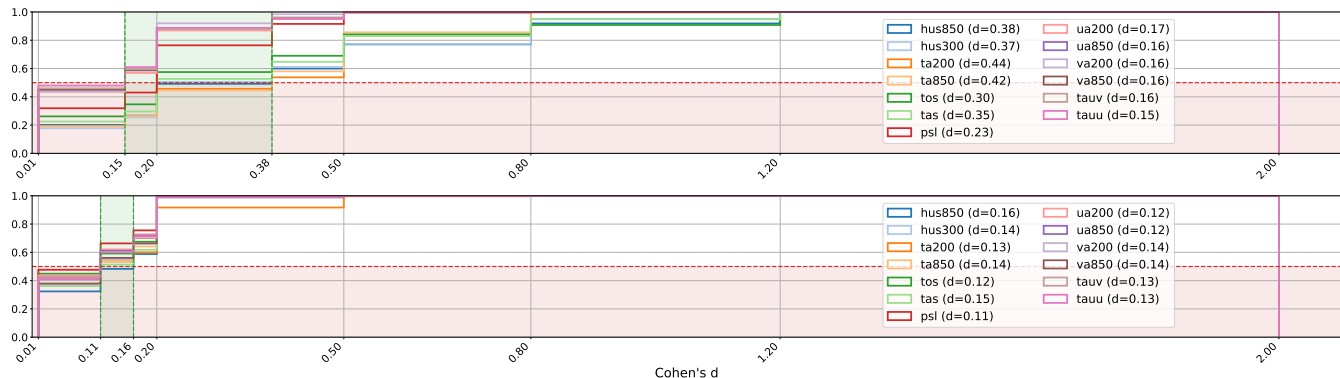

**Figure 6.** Cumulative Density Function (CDF) for the effect size between the two LENS2 sub ensemble, comparing the default CMIP6 BMB forcing to the smoothed forcing. We show the CDF for the reference period 1990-2014, where the forcing is different (top), and for the control period 1960-1989, where the forcing is the same (bottom). The values of $d$ in parentheses in the plot legends, list the respective median effect size for the variables.

used to validate statistical indistinguishability should be capable of resolving this margin of magnitudes. The categorization from Cohen remains reasonable also in the realm of climate simulations. We can set a threshold for small differences with an effect size of up to $d = 0.2$ to be tolerable, and only effect sizes larger than that to represent statistical differences in climatological states.

### 4.2.2 The scores and replicability test

In this section, we evaluate the scores from Section 3.1 and statistical tests from Section 3.4. It is insightful to look at the bootstrap distributions of the effect sizes that are generated by the various scores. In Figure 7 we show in red the distributions generated from the control, and in blue from the reference period. The text boxes in the plots list the quartiles (25%, 50%, and 75% quantiles) for the effect size calculated from the climatologies over the global domain, and the effect size $d$, calculated from the distribution of the single value scores. We can see that the effect size $d$, based on the scores, takes values somewhere within (0.0,0.17) for the control, and within (0.00,1.04) for the reference. However, taking always the highest value among all scores for a specific variable, the range becomes (0.07,0.17) for the control, and (0.19,1.04) for the reference. Taking the larger value range makes sense, when computing a combined score for the test, triggering the metric taking the largest value. We observe that the mean effect size after computing the scores is generally a bit larger than the mean effect size based on the grid point values of the climatologies. This is an advantage, as larger effect sizes can be resolved using fewer samples. Such an analysis, as above, can be used to estimate the sample size that we need to resolve the most subtle of differences between two samples. After determining the effect size and standard deviation using the bootstrap method, we can use a power analysis to calculate the required sample size.

A different method to compute the power for a given effect size and sample size is based on a Monte Carlo method and visualized in Figure 8, where we plot the power of the statistical tests presented in Section 3.4 against the effect size. We

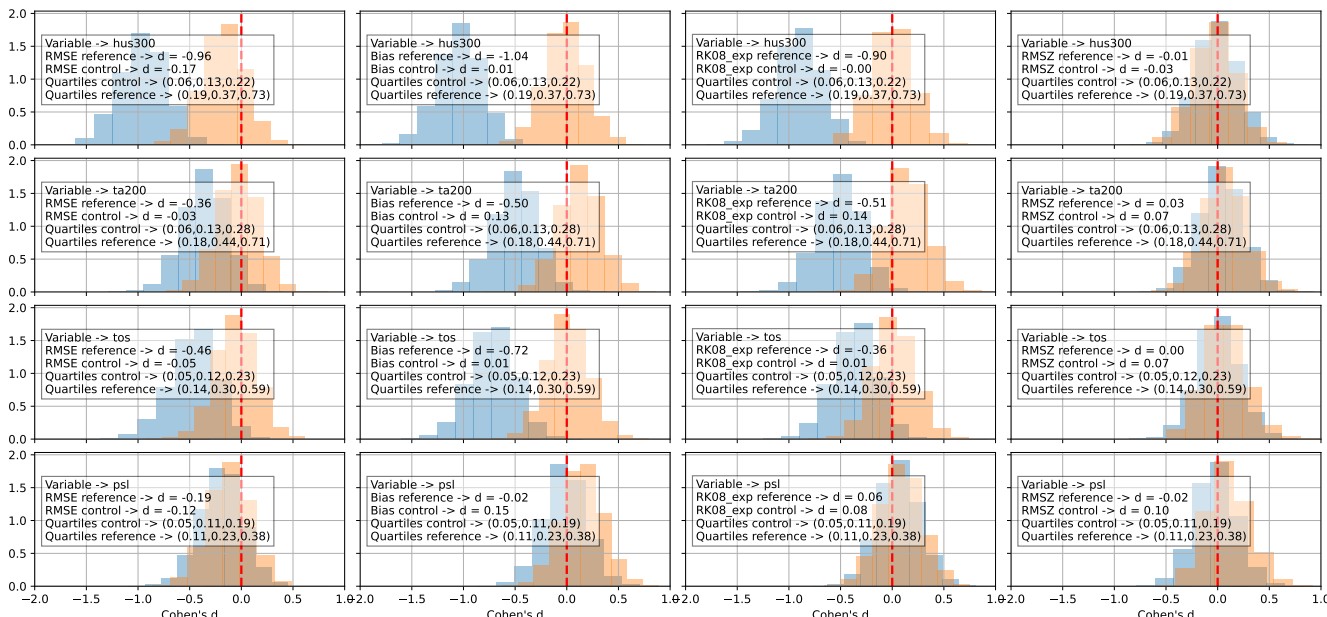

**Figure 7.** Bootstrap distributions for the effect sizes of the control period 1960-1989, where the BMB emission forcing is the same (in red), and for the reference period 1990-2014, where the forcing is different (in blue), for the variables hus300, ta200, tos, and psl. The boxes in the plots list the variable name, the effect sizes between the distributions of the respective scores for the control and reference periods, and the respective quartiles.

generate the data for those plots via repeated random sampling of datasets with ensemble sizes $n = 5, 10, 15, 20, 25, 50$. We compute the effect sizes from two randomly drawn samples via the bootstrap method. For each two samples, we apply all four statistical tests, and count the positive results (i.e., rejections of $H_0$). Before computing the power, we aggregate the effect sizes into bins with a width of 0.025. Finally, we compute the power, dividing the number of rejections by the total number of results per bin. We perform this for the Lorenz-96 toy model and for the LENS2 ensemble. Performing this with Lorenz-96 has the advantage that it is computationally not expensive, and we can draw from a large pool of ensemble members. The data for Lorenz-96 is generated using 100,000 random two-sample draws, for each sample size $n$, from a 21,000-member ensemble. The 21,000 member ensemble is just the joint ensemble containing the 1000-member ensembles that we generated for the 21 different values for $F$ (see Section 4.1.2). For the LENS2 ensemble, we generate the data with 7500 random draws from the full 100-member ensemble. It is likely that the results for the LENS2 ensemble are biased, as we probably undersample the distribution function with merely 100 members, and use fewer repetitions than for Lorenz-96. Both Lorenz-96 and LENS2 show fluctuations in the graphs, with fewer fluctuations for Lorenz-96. Yet, the results for both models seem compatible with each other to a large extent.

We can observe clear differences in performance among the statistical tests, while the test that performs best, for Lorenz-96 and LENS2, is the T-test, closely followed by the B-test. For the Lorenz-96 model, the bootstrap test seems superior for very

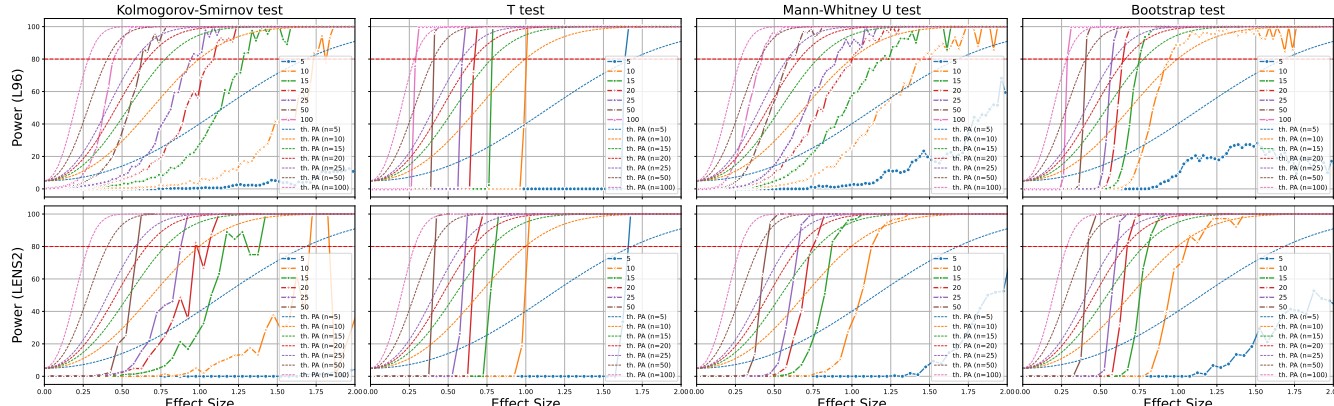

**Figure 8.** Monte Carlo power analysis of the dependency of the power on the effect size. The plots show the power of the four tests, T-test, U-test, KS-test, and B-test, for different ensemble sizes, and for two different models, the Lorenz-96 toy model and the data from the LENS2 ensemble.

small samples (20 or fewer members). This cannot be observed for the LENS2 ensemble. It is not clear if this is due to the fewer number of repetitions, or because of the higher complexity of the LENS2 datasets. We further observe that the effect of increasing the ensemble size becomes smaller towards smaller effect sizes. doubling the ensemble size from 5 to 10, leads to resolving effect sizes of $d = 1$ with 10 versus $d = 1.65$ with 5 members ($\Delta d = 0.65$). However, doubling the ensemble size from 25 to 50 members leads to $d = 0.4$ with 50 from $d = 0.6$ with 25 members ($\Delta d = 0.2$). In other words, it is getting harder to increase the resolution further. In general, doubling the ensemble size, seems to increase the resolution by approximately 30%. According to this, ensembles with 100 members would resolve effect sizes of $d = 0.25$.

### 4.2.3 Evaluating the Impact of the BMB Forcing on the LENS2 Ensemble

The CMIP6 prescribed biomass burning (BMB) emissions forcing, contains estimates of the emitted aerosol concentrations based on climate proxies such as sedimentary charcoal records, or records of reported wildfires, and data from the Global Fire Emissions Database (Van Marle et al., 2017). The latter is derived from satellite data during 1997-2016 (Van Der Werf et al., 2017). A consequence of the heterogeneous data sources are inconsistencies in the annual variability of the forcing. during the period from 1997 to 2016, a significant and sudden increase in annual variability of the BMB emissions is present (Fasullo et al., 2022; Van Marle et al., 2017). The CESM2 model shows high sensitivity to this abrupt change in variability, and reacts with an anomalous warming in the Northern Hemisphere in the extra-tropics. The warming trend is reduced again, once the variability of the emissions becomes more modest after 2016 (Fasullo et al., 2022). To weaken the impact of this anomaly, but still keep the CMIP6 prescribed forcing, 50 members of the LENS2 ensemble are forced with the original dataset, and 50 members are forced using a smoothed version, flattening the peaks in variability between 1997 and 2014.

In Figure 9, we show the 24-year ensemble-mean climatology (1990-2014) of the 2m air-temperature difference between the first and second 50 members of the LENS2 100-member ensemble. The first 50 members are forced with the original CMIP6

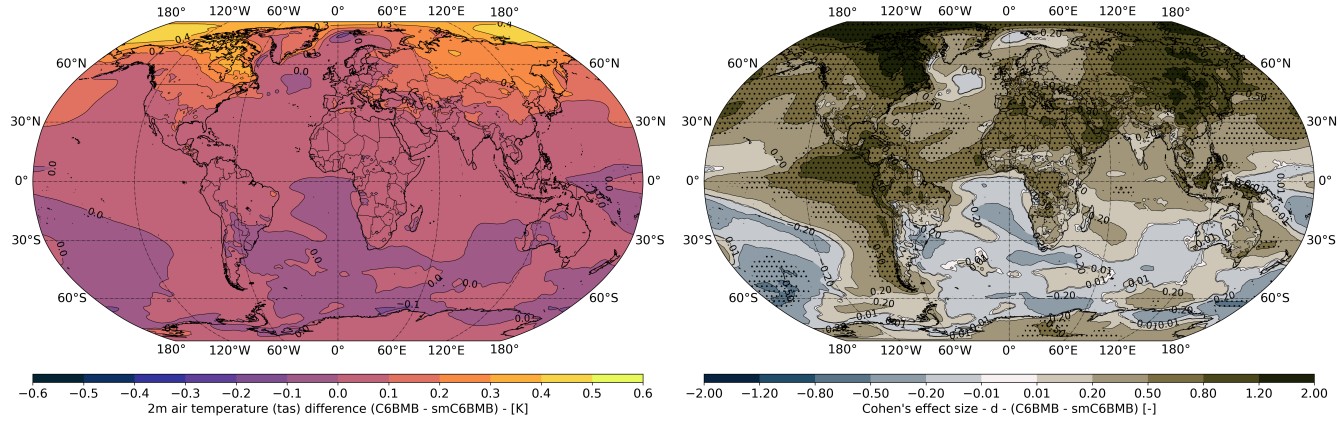

(a) ΔT - tas - for C6BMB minus smC6BMB climatology 1990-2014

(b) Effect size d - tas - for C6BMB/smC6BMB

**Figure 9.** (a) Differences in 2m air temperature (tas) between the smoothed (smC6BMB) and unchanged biomass burning (BMB) emissions forcing (C6BMB). The differences are taken between the respective 1990–2014 climatologies (C6BMB minus smC6BMB). (b) Effect size (Cohen's $d$) comparing the different BMB forcing. We additionally show the grid points where the differences are statistically significant, indicated by black dots.

BMB forcing (C6BMB), and the second 50 members are forced with the smoothed BMB CMIP6 forcing (smC6BMB). We can
see that there are significantly warmer temperatures for the members forced with C6BMB ($\Delta T > 0$). We observe a 0.2 to 0.5 K temperature difference in the extratropical regions, especially above Russia and Central Asia and parts of Northern America. Right to this, in Figure 9, we show the corresponding effect sizes and regions where the differences are significant, indicated with black dots. We computed the effect size from the two temperature samples at each grid point employing the bootstrap method. To identify the statistical significance of the differences, we applied a T-test at each grid point to the two ensembles,
where we set $\alpha = 0.05$. As we showed before, differences between two ensembles show small effect sizes of up to $d = 0.16$, even if they represent the same climate. Based on this, we have established the threshold for significant difference a bit higher at $d = 0.2$. In the figure, we have extended regions with effect sizes $d > 0.5$, $d > 0.8$, and even $d > 1.2$. Consequently, the differences in 2m air-temperatures originating in the use of the two different BMB forcing are clearly significant, and should be detected by the replicability test.
In Table 3, we list the results of the replicability test for 13 variables from the LENS2 ensemble. The variable meaning, units, and the corresponding observation datasets are listed in Table 1. The rows in Table 3 are sorted in decreasing order by the magnitude of the median effect size (in the table *median*). Where we first calculate the effect size, $d_i^X$, for variable $X$ at grid point $i$, between the two differently forced ensembles, and thereafter compute the median among all values $d_i^X$. Additionally, we list the median effect size calculated over the regions that we identified as significant using the T-test (in the table *median*
*SD*). The ratio of significant to non-significant differences is listed as a percentage in column *SD ratio*. The remaining columns list the effect sizes between the two samples, $d_S^X$, based on the scores, and the minimum p-value from the respective tests. We compute the $d_S^X$ using Equation 14 with $\mu_i$ being the ensemble average of the scores of the respective sample, and $\sigma$ the

**Table 3.** LENS2 replicability test for the period 1990-2014, comparing the default CMIP6 BMB forcing to the smoothed forcing. The *median* column lists the median effect size of two ensembles at the grid point values of the climatological fields of the variables. The *median (SD)* lists the median effect size of the climatological fields, only taking into account the grid points with significant differences. The column *SD* lists the amount of significant differences in percentage of the full number of grid points.

| var. | median | median$_{(SD)}$ | SD | $d_{eRMSE}$ | $d_{eBIAS}$ | $d_{eRK08}$ | $d_{comb}$ | $p_{min,tt}$ | $p_{min,ks}$ | $p_{min,mw}$ | $p_{min,bs}$ |
|---|---|---|---|---|---|---|---|---|---|---|---|
| ta200 | 0.443 | 0.694 | 52% | -0.35 | -0.495 | -0.529 | -0.625 | **0.003** | **0.006** | **0.003** | **0.003** |
| ta850 | 0.424 | 0.666 | 52% | -0.517 | 0.643 | 0.691 | 0.398 | **0.001** | **0.001** | **0.003** | **0.001** |
| hus850 | 0.385 | 0.766 | 48% | -0.86 | -0.807 | -0.749 | -0.924 | **0.000** | **0.000** | **0.000** | **0.000** |
| hus300 | 0.375 | 0.778 | 46% | -0.937 | -1.028 | -0.881 | -1.081 | **0.000** | **0.000** | **0.000** | **0.000** |
| tas | 0.354 | 0.693 | 44% | -0.522 | -0.997 | -0.818 | -1.022 | **0.000** | **0.001** | **0.000** | **0.000** |
| tos | 0.304 | 0.7 | 27% | -0.442 | -0.706 | -0.722 | -0.882 | **0.000** | **0.001** | **0.000** | **0.000** |
| psl | 0.231 | 0.481 | 19% | -0.188 | -0.022 | 0.024 | -0.127 | 0.358 | 0.068 | 0.216 | 0.351 |
| ua200 | 0.166 | 0.489 | 11% | -0.297 | -0.234 | -0.25 | -0.337 | 0.099 | 0.396 | 0.100 | 0.093 |
| va200 | 0.165 | 0.461 | 6% | 0.006 | 0.046 | 0.135 | 0.071 | 0.510 | 0.179 | 0.454 | 0.505 |
| ua850 | 0.16 | 0.484 | 9% | 0.064 | 0.041 | 0.076 | 0.087 | 0.671 | 0.272 | 0.467 | 0.651 |
| va850 | 0.16 | 0.484 | 9% | -0.156 | 0.316 | 0.367 | 0.28 | 0.077 | 0.112 | 0.083 | 0.076 |
| tauv | 0.157 | 0.484 | 10% | 0.279 | 0.042 | 0.09 | 0.21 | 0.170 | 0.068 | 0.216 | 0.165 |
| tauu | 0.149 | 0.503 | 9% | 0.114 | -0.171 | -0.186 | -0.037 | 0.360 | 0.272 | 0.278 | 0.353 |

pooled standard deviation of both samples. We reject the null hypothesis as soon as one combination of test and score leads to a minimum $\tilde{p}$-value (see Equation 2) that is below the significance level $\alpha = 5\%$. Consequently, the variables hus850, hus300, ta850, ta200, tas, and tos, are identified by the test as non-replicable (in bold). When comparing the median from the table to the median of the effect sizes from the control, we can see that the test performs reasonably well. From comparing with Figure 8, we expect that we can resolve effect sizes of about 0.38-0.4 with 50 members. In 3, we can see that the median effect size of the scores (columns 5-8) is always a bit higher than the median effect size calculated from the variable values at the grid points (column 2). Considering only the highest median effect size based on the scores (i.e., $\max_{\forall s \in S}(d_s^X)$) for each variable, and only variables that have a median effect size based on the variable values at each grid point, larger than 0.2 (i.e., ta200, ta850, hus850, hus300, tas, tos, and psl), we can see that we the median effect sizes for the scores are generally larger, with a minimum of $d = 0.188$ for psl, and the next smallest value, $d = 0.628$, for ta200. While the effect sizes are generally larger, the test still fails to reject psl, which seems to show significant differences as well (compare Figure 6). To reliably identify psl as non-replicable, we could do a power analysis to identify the sample size that is required for the given effect size. When comparing the CDF of psl between the control and reference period, the red lines in Figure 6, we would attribute it to the set of variables that show significant differences. In a context of climate replicability due to a technical change, it is typically enough to identify one variable to be non-replicable. Hence, for that use case, the test is robust, as it is likely that several variables are affected if the climate in non-replicable. As a visual aid and for comparison with the table, we show the spatial distribution

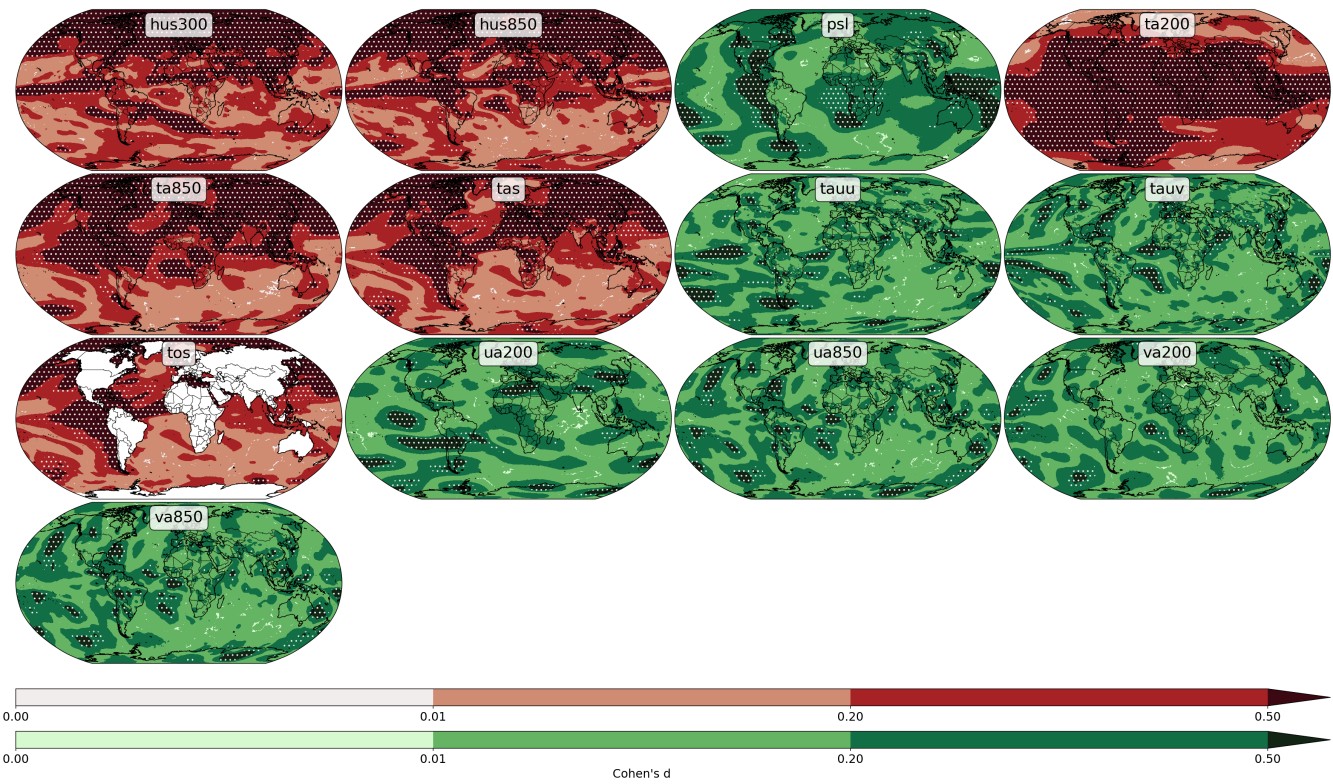

**Figure 10.** Spatial maps of effect sizes between the two LENS2 sub ensembles, comparing the default CMIP6 BMB forcing to the smoothed forcing. The maps additionally show the regions where the differences are significant, indicated with white dots. The color of the plots indicates the results of the replicability test for the variable. Plots in red indicate a rejection.

of the effect sizes for all the variables of the test. The plots in Figure 10 are masked, and only show effect sizes in regions
where the differences are significant. Further, we only resolve effect sizes below 0.2, between 0.2 and 0.5, and above 0.5. For variables that were rejected by the test, the maps show shaded areas in red, and for the variables that were not rejected, the areas are shaded in green.

### 4.2.4    Evaluating the Impact of the AMOC current for initializing the LENS2 Ensemble

The LENS2 historical simulation starts from initial states generated from a long pre-historical control simulation. In Table 2, we
list the characteristics of the LENS2 ensemble that are important for our evaluation. Members 11-20 and 51-60 are initialized from an AMOC phase showing a rather strong current (IC1231). Members 31-40 and 71-80 are initialized from an AMOC phase showing a rather weak current (IC1281). The different BMB forcing imposed on the first and second half of the ensemble only affects periods much later in the simulation. The forcing is identical for all members of the ensemble during the period 1850 to 1997. Therefore, statistical differences between the differently initialized sub-ensembles, can be attributed to the

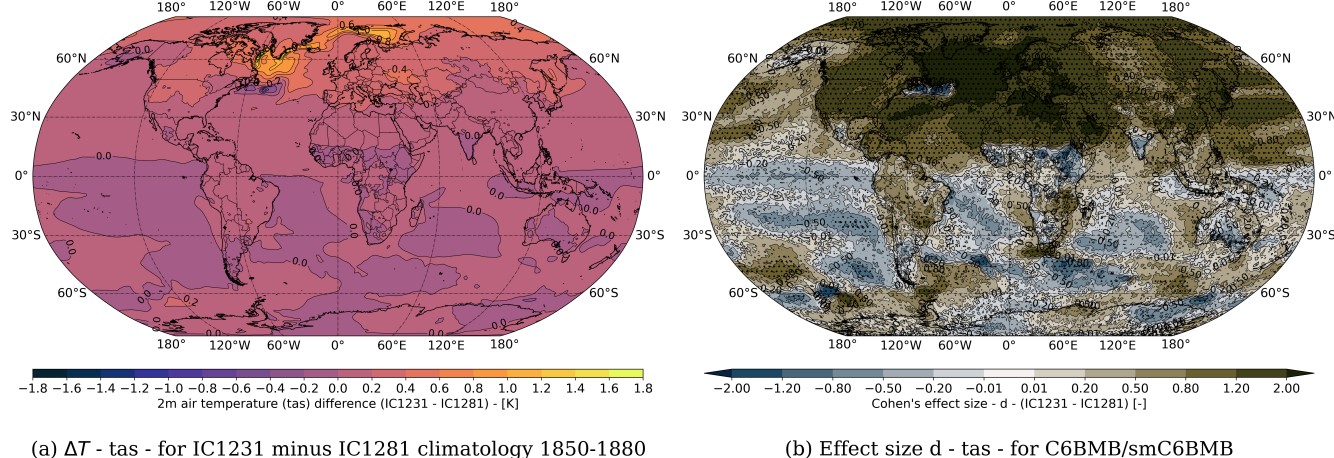

(a) $\Delta T$ - tas - for IC1231 minus IC1281 climatology 1850-1880

(b) Effect size d - tas - for C6BMB/smC6BMB

**Figure 11.** (a) Differences in 2m air temperature (tas) between the simulations starting from an AMOC phase with a strong current (IC1231) and an AMOC phase with a weak current (IC1281). The differences are taken between the respective 1850–1880 climatologies (IC1231 minus IC1281). (b) Effect size (Cohen's $d$) comparing the differently initialized simulations. We additionally show the grid points where the differences are statistically significant, indicated by black dots.

different AMOC phases, if considering periods in the first few decades of the simulation. Similarly, as for the comparison in the previous section, we plot the difference of the climatologies and the effect sizes over the global domain for comparing the projections of the two sub-ensembles in Figure 11. The reference period is 1850-1880. We observe large temperature differences in the Northern Hemisphere, that are especially pronounced in the Labrador and Norwegian Sea, ranging from $\Delta T = 0.2$ to $\Delta T = 1.0$K. Furthermore, the effect size is above 0.5 for almost the whole region in the Northern Hemisphere,

and lies above 1.2 and 2.0 for large regions. Consequently, the climate is significantly different in the first 30 years when starting from two very different AMOC states. A warmer climate in case of a stronger AMOC current is not surprising, as the AMOC current transports heat from the southern Atlantic northwards. A stronger current can be expected to bring more heat to the northern Atlantic than a weaker current would do. Meccia et al. have observed the same, investigating the multi-centennial variability of the AMOC (Meccia et al.).

An important difference to the previous case, where we compared two sub-ensembles with 50 members each, is that we are now comparing two sub-ensembles with only 20 members each. This has consequences on the accuracy of the test. In Figure 8 we can see that with 20 members, using the T-test, we can resolve effect sizes of about 0.6 to 0.7. This is reflected in the results that we list in Table 4. The test detects non-replicability for the variables with scores above $d = 0.6$. The vertically resolved temperature at pressure level 200hPa (ta200), shows an effect size of a maximum value $d = 0.546$, and is not detected

by the test. The scores again show effect sizes typically a bit larger than the median from the effect sizes on the global domain. Considering the grid point based values, we can resolve effect sizes of above 0.35, which is a reasonable performance using only 20 members. Yet, the performance of the test in this case is clearly not as good as in the previous case, using 50 members. Further, in Table 4, we can see that only the T-test and B-test correctly reject the null hypothesis for some of the variables.

**Table 4.** Results of the replicability test for the two LENS2 sub ensembles starting from different AMOC phases. The *median* column lists the median effect size of two ensembles at the grid point values of the climatological fields of the variables. The *median (SD)* lists the median effect size of the climatological fields, only taking into account the grid points with significant differences. The column *SD* lists the amount of significant differences in percentage of the full number of grid points.

| var. | median | median$_{(SD)}$ | SD | $d_{eRMSE}$ | $d_{eBIAS}$ | $d_{eRK08}$ | $d_{comb}$ | $p_{min,tt}$ | $p_{min,ks}$ | $p_{min,mw}$ | $p_{min,bs}$ |
|------|--------|-----------------|-----|-------------|-------------|-------------|------------|--------------|--------------|--------------|--------------|
| tas | 0.525 | 1.33 | 41% | -0.016 | -0.721 | -0.816 | -0.663 | **0.018** | 0.081 | 0.086 | **0.007** |
| hus850 | 0.486 | 1.259 | 41% | -0.296 | -0.683 | -0.539 | -0.638 | **0.045** | 0.081 | 0.120 | **0.032** |
| ta850 | 0.472 | 1.279 | 41% | 0.265 | 0.658 | 0.559 | 0.743 | **0.028** | 0.175 | 0.064 | **0.020** |
| hus300 | 0.364 | 1.116 | 35% | -0.141 | 0.808 | 0.63 | 0.781 | **0.018** | 0.175 | **0.034** | **0.005** |
| ta200 | 0.358 | 0.904 | 17% | -0.018 | 0.546 | 0.508 | 0.432 | 0.105 | 0.336 | 0.208 | 0.086 |
| psl | 0.314 | 0.911 | 18% | -0.172 | -0.368 | -0.307 | -0.382 | 0.248 | 0.336 | 0.199 | 0.248 |
| va850 | 0.25 | 0.852 | 8% | 0.521 | -0.075 | -0.089 | 0.001 | 0.117 | 0.175 | 0.064 | 0.116 |
| ua850 | 0.238 | 0.826 | 6% | 0.157 | -0.042 | -0.06 | 0.017 | 0.633 | 0.832 | 0.543 | 0.622 |
| va200 | 0.231 | 0.798 | 4% | 0.508 | 0.11 | 0 | 0.141 | 0.129 | 0.336 | 0.114 | 0.133 |
| ua200 | 0.228 | 0.876 | 11% | 0.124 | 0.304 | 0.134 | 0.271 | 0.356 | 0.571 | 0.337 | 0.350 |

None of the other tests, except the U-test in case of the specific humidity at pressure level 300 hPa (hus300), trigger a rejection. This shows the benefit of our methodology, that employs several tests and metrics as the basis for a rejection. However, a look at the data and the plots of the other variables, suggests that significant differences are present as well for ta200, psl, and ua200. For instance, comparing the mean effect size from the respective three variables, taking only the grid points with significant differences into account, we can see that variables in the previous section with a similar order of magnitude of the mean, have been rejected. The visual assessment of the spatial plots in Figure 12 suggests the same conclusion.

## 5 Conclusions

We proposed and conducted a thorough analysis of ensemble-based replicability tests. Additionally, we developed and presented a new replicability test. While our test builds upon previously developed methodologies, to the best of our knowledge, those lacked a rigorous procedure for assessing their performance and effectiveness in distinguishing ESM outputs that represent different climates. To address this shortcoming, we proposed using Cohen's $d$ effect size to objectively quantify the statistical differences between two ensembles of simulations, providing a precise method to quantify what constitutes *different climates*. Following Cohen's framework for classifying differences between ensembles according to the effect size $d$, we established that a threshold of $|d| = 0.2$ is a reliable criterion to define the statistical distinguishability between two ensembles of simulations. Hence, a trustworthy replicability test should, at minimum, be able to detect differences of $|d| \geq 0.2$. The possibility of increasing this threshold to $|d| = 0.35$ seems reasonable, depending on the desired degree of accuracy and purpose of

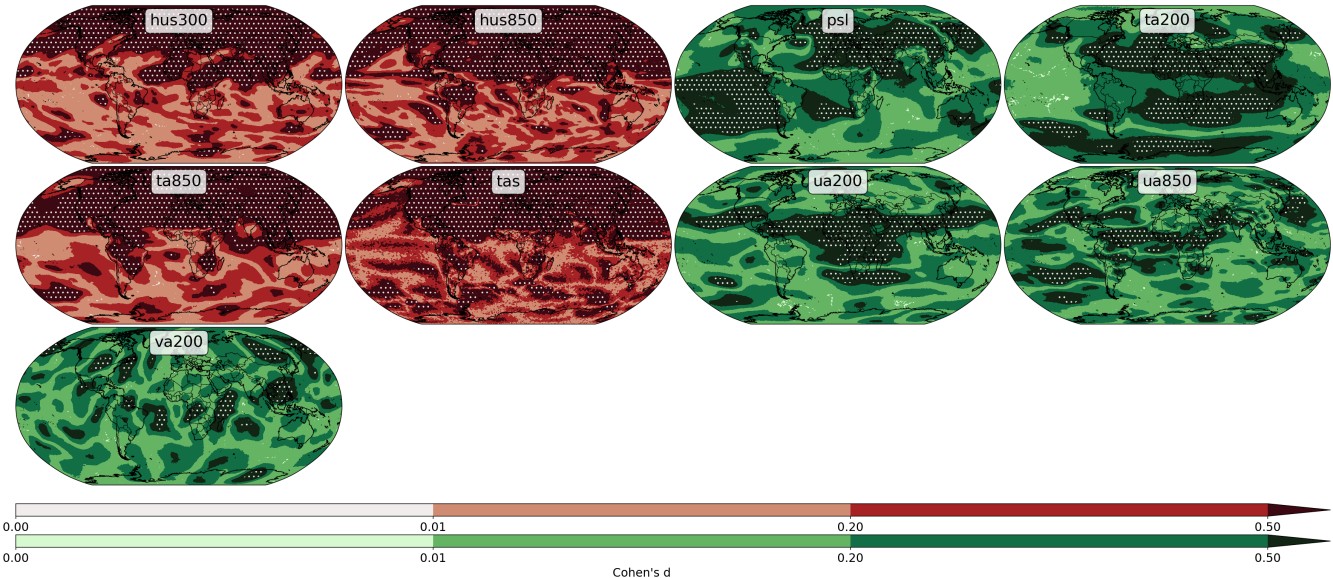

**Figure 12.** Spatial maps of effect sizes between the two LENS2 sub ensembles starting from different AMOC phases. The maps additionally show the regions where the differences are significant, indicated with white dots. The color of the plots indicates the results of the replicability test for the variable. Plots in red indicate a rejection.

the replicability test. Nevertheless, providing a comprehensive list of thresholds requires a case-by-case scientifically guided analysis and falls outside the scope of this work.

As mentioned, the structure of the test resembles that presented in (Massonnet et al., 2020), however, we suggested two key adjustments to improve the performance of the test. First, we included two additional metrics to the test, the RMSE and Bias, and unified the value range of the scores to (0,1) enabling the calculation of a third score, the combined score of the three scores of the test. Secondly, we added three additional statistical tests to the replicability methodology, including commonly used non-parametric tests, the T-test, U-test, and KS-test, and developed a new test based on the bootstrap method, the B-test. In contrast to a single score and statistical test, we define the outcome of the test by the minimum $p$-value resulting from all combinations of tests and scores.

We assessed the performance of our methodology using the FPR and the power. Employing normally distributed samples and the Lorenz-96 toy model, we demonstrated that achieving a high power aiming to resolve the established threshold of $d = 0.2$ requires ensembles with more than 50 members. Our analysis revealed that one would need ensembles of more than 100 members to detect effect sizes of $d \sim 0.4$ with 80% confidence, and 50 members for effect sizes of $d \sim 0.8$. However, we observed that the effect sizes comparing the scores rather than the climatological fields at the gridpoint basis, is generally larger. Therefore, the threshold of $d = 0.2$ based on the grid point basis, translates to a larger value after computing the scores. For instance, our analysis based on the LENS2 ensemble shows, that for the sea surface temperature (tos), the median effect size of $d \geq 0.3$ at the grid points, translates to a value of $d \approx 0.78$ with the combined score, and can then be resolved using 50

members (see Table 3). Our methodology rejects all variables with $|d| > 0.3$. It only failed to detect the differences in sea level pressure, the only variable that displayed a lower effect size but still above the threshold ($0.2 < |d| < 0.3$). When analyzing the first 30 simulated years of 20-member ensembles generated with different initial conditions, following two extreme phases of the AMOC, our methodology detected effect sizes of $|d| > 0.35$, which is slightly higher as for the case with 50 members. However, if the aim is to just test for climate replicability, using 20 member ensembles might be sufficient, as it is enough to have a single rejection present for all variables under consideration.

Both the revised replicability methodology we have presented, and the objective measure of statistical difference given by Cohen's effect size that we introduced, will be beneficial for future phases of CMIP and other climate model intercomparison projects. It will increase the confidence in the statements derived from simulations that are performed on different clusters with the same model, and it will effectively increase the value of such simulations. With our methodology, we can detect inconsistencies up to any degree of accuracy. Therefore, it can be applied to detect and entirely remove artificial contributions to climate signals in our simulations introduced by differences in computing environments.

The requirement of sufficient ensemble sizes has been formulated before (Mahajan et al., 2017; Baker et al., 2015; Zeman and Schär, 2022; Milroy et al., 2018). While the ensemble tests from Milroy et al. (Milroy et al., 2018) and Zeman et al. (Zeman and Schär, 2022) provide viable methodologies testing for acceptable climates of isolated atmosphere and land models, it is unclear how to provide cost-efficient tests for statistical consistency in long climate simulations with coupled models, including the slower evolving processes of the ocean component. In future research efforts, this matter needs to be addressed.

*Code and data availability.* The code to reproduce the statistical tests and figures is archived at Zenodo under 10.5281/zenodo.15052430 (Keller and Alerany Solé, 2025). The LENS2 data for the evaluation is available as lossy compressed archives at Zenodo as well. The 100 member ensembles for three periods are available for 1850-1880 under 10.5281/zenodo.15049719 (Alerany Solé and Keller, 2025c), 1960-1990 under 10.5281/zenodo.15049671 (Alerany Solé and Keller, 2025b), and 1990-2014 under 10.5281/zenodo.15045355 (Alerany Solé and Keller, 2025a). The uncompressed data can be requested by contacting the corresponding author.

*Author contributions.* KRK and MAS developed the replicability methodology presented in this article. The evaluation of the methodology based on the toy models was performed by MAS. The evaluation based on the LENS2 ensemble was performed by KRK. The preparation of the original draft was led by KRK, while MAS substantially contributed to writing it. MA supported the writing of the original draft. Finally, all authors contributed to revising the draft and provided comments to improve it.

*Competing interests.* The contact author has declared that none of the authors has any competing interests.

*Disclaimer.* The writing of this article has been performed entirely without the use of Large Language Models such as ChatGPT or similar.

*Acknowledgements.* This work is based on the Hpc AlliaNce for Applications and supercoMputing Innovation (HANAMI) project funded by the European High Performance Computing Joint Undertaking (EuroHPC JU) under the European Union's Horizon Europe framework program for research and innovation and Grant Agreement No. 101136269. Views and opinions expressed are, however, those of the author(s) only and do not necessarily reflect those of the European Union or EuroHPC Joint Undertaking. Neither the European Union nor the granting authority can be held responsible for them. We acknowledge the support from the Barcelona Supercomputing Center (BSC) by providing internal resources from the Earth Science department on the Marenostrum 5 supercomputer, to perform our analyses. Further, we are grateful to the Data and Diagnostics team at the BSC Earth Science department to provide us with the ensemble data from the LENS2 ensemble and the observational datasets used in this work. We further acknowledge the CESM2 LENS2 project and the supercomputing resources provided to it by the IBS Center for Climate Physics in South Korea.

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
