# Peer review of "Replicability in Earth System Models"

_EGUsphere, 2025_

## Author Response (AR1)

Foremost, we want to take a moment to express our gratitude for the time and effort the reviewers put into evaluating and commenting our manuscript. It has helped us to better understand certain concepts and to (hopefully) improve the clarity of the statements in our article.

Before responding to the specific comments, we would like to expand on the differences between our methodology and other methodologies in the literature. The majority of replicability tests in the literature are based on individual model components and aim for acceptable model climates after a technical update. The works from Baker et al. target the statistical consistency of isolated atmosphere and ocean models. Likewise, Mahajan et al. consider isolated ocean models, Zeeman et al. consider regional atmospheric models, and Milroy and Price-Broncucia consider atmosphere and land models. These methodologies always consider short simulation lengths and fixed forcings, therefore, do not address interannual climate variability. It is credible that such tests are capable of detecting significant changes in the component's climate after a model update. In addition, tests for continuous integration and model development are subject to stringent requirements on cost and time-to-completion. Therefore, performing long simulations with large ensembles is not practicable. However, in the context of comparing long climate simulations with fully coupled models, that exceed the period for climate predictability, for instance based on ENSO and AMOC, testing isolated components individually is not sufficient. By performing short simulations with forced models, we miss the dynamic feedbacks between the different components, and we cannot validate the statistics of slowly evolving cyclic events. In fact, taking into account the different time scales of atmosphere and ocean processes, both Mahajan and Baker acknowledge that a minimum of 1 year of simulation is required to evaluate the statistical consistency of the ocean processes. In addition, Baker notes in her work on the POP-ECT ocean model consistency test, that it should not be used for evaluating statistical consistency of the interannual variability of events such as the ENSO (Evaluating statistical consistency in the ocean model component of the Community Earth System Model, Section 6, 2016).

Response inline with the Reviewer's comments:

**Reviewer 1**

Comment 1:

The article discusses replicability from a statistical and computational perspective, focusing on the comparison of results between different environments. However, in order to be able to replicate an experiment in a robust way and to properly adjust the parameters to the available hardware, it is also essential to know how the software is implemented. Do you think that accessibility to the model code should be part of the replicability criteria or not?

Response 1:

When replicating a model climate, it is essential to reproduce the individual steps taken to acquire the original climate. However, our methodology represents a generic tool to detect

inconsistencies in climates. It is certainly recommended to use workflow managers and to follow the FAIR principles when setting up replicability experiments. And it is unlikely to reproduce a model climate when those principles are not respected. However, although access to the code would help understand the discrepancies found when running our test, we do not see it as a requirement for it.

Comment 2:

You mention several essential technical and methodological requirements to ensure replicability, such as the definition of the computational environment, the minimum size of the ensembles or the use of robust metrics such as the Z-score. Along these lines, do you consider it useful to complement this methodology with a guide to good documentation practices? For example, include detailed descriptions of experiments, input parameters, model configuration, hardware characteristics, and establish one or more 'base cases' to serve as a reference to validate that the model is correctly configured and responds as expected.

Response 2:

While we agree that a guide for good documentation practices is essential, and that it is desirable to follow such guidelines in the climate community, we think that it is attached to a slightly different topic. There exist recommendations on the categorization of reproducibility, replicability, and repeatability. In the context of replicability, we do not have any requirement on the generation of the model climate. A replicability methodology identifies differences in climates, without the knowledge on how the climate was generated. Experiment documentation on input parameters and experiment configuration, etc. is part of what is typically attributed to model reproducibility and repeatability. In order to reproduce or repeat an experiment, for instance, from a different modelling centre, this documentation is required. Only then one is able to reproduce the steps that have been performed in the original experiment.

Comment 3:

You propose a threshold of d ≤ 0.2 as a criterion for considering two simulations to be statistically indistinguishable. Do you think this threshold can be generally applied to any model, variable or time scale? Or do you think it should be adapted according to the type of variable (e.g. temperature versus precipitation) or the geographic region modelled?

Response 3:

This is an interesting question, but unfortunately lies outside the scope of this article. We have no evidence from other models yet, however, it is conceivable that the effect size is different for other models and time scales. One would need to do a study including a variety of models that provide long climate simulations with large ensembles. We certainly intend to

investigate this in future works. Our analysis shows that at least within the LENS2 ensemble and for the variables that we have tested, the effect size is quite consistent with a median between 0.11 and 0.16 for all variables (see Figure 6, second plot).

Comment 4:

The study is based on CESM2 and the Lorenz-96 model as a conceptual test. Have you considered how this methodology would perform in very high resolution models (below 10 km) or with more complex couplings, such as those including biogeochemical or cryosphere components?

Response 4:

Extending methodologies such as the ones we present to high resolution is challenging due to the increased computational costs. However, we believe that the basic idea underlying the test would still be valid. It remains an open question whether the ensemble size and simulation length is affected, aiming to achieve the same power and FPR as for low resolution models. Certainly an important question that needs to be addressed in future works.

Comment 5:

Your methodology focuses on evaluating replicability through statistical comparison of model outputs across computing environments. Do you think that combining this approach with code-level analysis tools (e.g., static code checkers as FortranAnalyser) could further enhance replicability? Specifically, could such integration help detect unintended changes in the model's behavior introduced during code development, and thus complement your statistical framework?

Response 5:

Complying to good programming practices is essential for reducing unintended behavior during the runtime. It is good advice to regularly check the code base using tools as you suggest. There have been cases of non-replicability due to false initialization of fields in EC-Earth3 running on 2 different supercomputers, as reported in DOI:10.5194/gmd-13-1165-2020. In the context of replicability test for continuous integration, such tools can indeed provide valuable information to identify the sources for non-replicability. Therefore, it represents a beneficial addition for such tests. However, we rather see it as a component supporting the continuous development suite than as a part of the replicability test itself.

**Reviewer 2**

**General**

Comment 1:

I want to make the authors aware of this recently published work in GMD, which is a follow on to the MIlroy et al. 2018 work and advocates short ensembles:

"The ensemble consistency test: from CESM to MPAS and beyond" by Teo Price-Broncucia et al. (https://gmd.copernicus.org/articles/18/2349/2025/)

Response 1:

Thank you for providing us with this resource. In fact, we have been thinking on implementing the methodology from Baker et al. to compare it to ours. This article will help us in doing this. Further, the methodology to identify correlated variables seems interesting and could improve our replicability test as well. We further added it as a reference in the revised document (Line 115).

Comment 2a:

I would like to see more discussion and experiments on the length of the ensemble needed for this approach in practice.  The motivation section put a lot of emphasis on the possibility of missing short-term effects or not noticing that short-term effects (like shocks) might go away later if the ensemble length is not sufficient.   I don't agree with the general statement that short-term effects are *not* relevant for multi-decadal simulation, and that one has to have a long simulation to make such a determination.  In particular, this argument goes against the findings in Milroy 2018, which shows that the signatures of later differences are already present earlier in the runs (after several time steps).  The authors cite papers such as Teng et al. 2017 in their argument that long runs are needed, but that study is using monthly averages, which is much longer than Milroy or Price-Broncucia papers suggest.  Do we know what these runs with initial shocks look like after a few timesteps? For the contributions here to be meaningful, I believe the authors need to test some of their longer runs and verify whether or not they can see the signal earlier. If the authors use the argument that a longer run is needed as a motivation, then it should be investigated/discussed more thoroughly. Also I don't believe they define how long of a run  is needed, which is a consideration for computational cost.   I think discussing cost is important as it is used for the motivation in this work.

Response 2a:

It is true that we do not provide an assessment of the required simulation length for testing the statistical consistency in case of a fully coupled model. We fully agree that this question needs to be addressed in future works, and we added a paragraph on this in the article (lines 157-160), however, a comprehensive assessment on the required length unfortunately lies outside the scope of the article.

We fully agree that a discussion on the cost is necessary. One could alleviate the overall cost by separating the methodologies into different fields of application. For instance, methodologies

from Baker, Milroy, Mahajan, and Zeeman, address the statistical consistency of the individual model components during the model development. Those tests are required frequently and need to be cost-efficient for practical reasons. On the other hand, methodologies such as we present in our article, and previously presented by Massonnet, aim to evaluate if two climatologies, produced with fully coupled ESMs in multi-decadal simulations subject to the same scientific constraints, are statistically identical. Such tests do not need to be performed as frequently. They need to be applied only when we want to compare different climate scenarios generated by the same model under different technical constraints (e.g., simulations on two different clusters).

Finally, after reconsideration, we fully agree with the reviewer that indeed short-term effects are relevant for long climate integrations, and we removed this statement entirely in the revised manuscript.

Comment 2b:

Another related point is that the Baker, Milroy and Price-Broncucia works are all using PCA to identify the signal earlier. Differences can manifest themselves in ways other than in mean differences (e.g., there could be cases where relationships between variables change, but not their means). I don't think that the effect size captures differences in variable relationships, as it is capturing the mean shift in the variables.

Response 2b:

We base the choice of variables on a study investigating variable dependencies using a cluster analysis (DOI:10.2151/jmsj.2012-A04 and DOI:10.1175/2011JAMC2643.1). By covering a broad selection of climate variables from the different clusters, we should be able to reveal issues that are caused from dependent variables. Including dependent variables is unfavorable, as it can lead to higher false positive rates (this is a consequence of the statistical nature of the test), however, it does not affect the statistical power of the test.

Comment 3:

As far as computational cost, another aspect of this work that is not clarified is the potential downside to comparing two large ensembles. For example, this two-ensemble approach is not ideal in the setting where users generally have much less compute power than the HPC centers developing the models. Also, it is generally not efficient if you want to quickly test changes against an established setting. This manuscript doesn't make that clear, and I think it should be given that the Baker, Milroy and Price-Broncucia papers avoid that.

Response 3:

We agree that additional work is required to improve the feasibility of the presented methodology. For instance, it is not clear how to extend it to high-resolution coupled models. We are currently exploring a variety of directions, including methods from machine learning to

reducing the ensemble sizes required for the test. Using a smaller sample size for the second ensemble is interesting, but generally requires the size of the first ensemble to be increased, as smaller ensemble sizes lead to a reduced statistical power. Similarly as for the simulation length, we agree that this should be explored in a future work, however, unfortunately does not lie within the scope of this article.

Comment 4:

Figures: The text in almost all of the figures is too small, especially the axes labels and legend text. Please either make the text larger or make the figures larger (many don't fill the width of the page) --or some combination. For example, Figures 2-12.

Response 4:

We took the suggestion from the template of 12cm width for two-column figures. However, after revising the manuscript preparation rules, there is nothing that constraints the width of the figures. Therefore, we extended the figures to the width of the document. In case the text size is still too small, we will revise it for the final version.

Comment 5:

The original CESM-LENS ensemble could be interesting to look at with this approach because most of the runs were done on a NCAR computer, but some were done at University of Toronto. And, if you look at the "known issues" for Sept. 2016 (https://www.cesm.ucar.edu/community-projects/lens/known-issues), there is a discussion on how someone discovered a systematic difference between the runs done at the two institutions. (I am not suggesting this as an addition for the present manuscript, but just making the authors aware in case it interests them).

Response 5:

Very useful indeed! This will be an interesting use case for us. Thank you very much for the hint.

**Specific**

–Line 114: Probably should add Milroy 2018 to this list of references

Response: The reference has been included in our updated version of the manuscript

–Line 124: Re: "one year or shorter". The runs in Milroy 2018 are much shorter than a year - only 9 timesteps. This should probably be mentioned here or in line 142. The reason is that later (line 146), it is stated that a 100 member ensemble is "cost prohibitive" at high-resolution. While I agree that may be true for a one-year run, I don't agree with that statement for a run that takes only 9 timesteps, for example, which appears to be the motivation for Milroy 2018 and then the more recent Price-Broncucia paper.

Response: We agree that this is not fair, given that the shorter test from Milroy et al. can indeed replace the CAM-ECT test from Baker et al. We have addressed this in the revised manuscript (Lines 130-132, 153-155, and 655-656).

–Line 125-139: As discussed above in the general comments, I don't agree with everything in this paragraph. If the authors use the argument that a longer run is needed as a motivation for their work, then it should be investigated/discussed more thoroughly with additional experiments. Then some discussion of cost is needed for the tradeoff between a long run and fewer ensemble members versus a shorter run and more ensemble members.

Response: We agree that this paragraph is misleading in the original document. Further, some of the statements are not supported by experimental evidence, as you mention. Therefore, we have removed it entirely from the manuscript. Instead, we have almost completely rewritten the motivation from line 123 to 169 to make our point on why we need long ensemble simulations, and why our methodology is a complementary contribution to the existing methodologies.

–Line 295: d is first mentioned here, but not defined until line 302 (and then more detail in line 312 and eqn 15). Could be better to define d closer to its first use.

Response: Thank you for this comment. Indeed, it makes more sense to introduce the concepts of false positive rate (FPR), power, and effect size before describing the statistical tests. We revised the text in section 3.3 (3.4 in prior manuscript version) and interchanged the two sections (see lines 282-309 in the revised manuscript).

–Line 530 (and others): It is not entirely clear to me how the threshold for significant difference (the suggested .2) would vary for other models or other variables or how exactly one would know that it needed adjusting. (Maybe this is an avenue for future work, but if so, please state.) The discussion on page 23 about what the procedure is for resolving the effect sizes was not totally clear to me.

Response: This is certainly an interesting question, and we want to address it in the future. We have answered a very similar comment from another reviewer. Therefore, we will quote our response here:

"We have no evidence from other models yet, however, it is conceivable that the effect size is different for other models and time scales. One would need to do a study including a variety of models that provide long climate simulations with large ensembles. We certainly intend to investigate this in future works. Our analysis shows that at least within the LENS2 ensemble and for the variables that we have tested, the effect size is quite consistent with a median between 0.11 and 0.16 for all variables."

We further indicated in the revised manuscript that when using a different climate model, those thresholds for the effect size might change (lines 181-183). We further revised the text on page 23, to better explain the concepts (lines 555-577).

–Line 630 - 634:  I think using the work in Baker 2015 to say that the CESM ensemble approach is too expensive is outdated, as the Milroy 2018 paper basically says that about the 2015 paper and improves on the CESM approach by proposing to use only 9 timestep runs.  Then the Price-Broncucia paper furthers the effectiveness of short runs.  For such short runs, it is harder to argue that larger ensembles are cost prohibitive even for high-res runs.

Response: Indeed. We have revised the document accordingly in the motivation section (lines 130-132 and 153-155) and in the conclusions (lines 655-656).

**Minor**

–Line 162: "statistically indifferent" is awkward.  Do you mean "statistically indistinguishable"

Response: Thank you, we have changed it to statistically indistinguishable

–Line 213: "the period for the climatology" ->"the time period for the climatology"

Response: We have changed it to what you are suggesting

–equation (11): should there be a bar over x_mi since it's an average like y_i?

Response: Indeed. We have updated the appearances in the revised manuscript, and indicate the time averages now with a bar over x_mi.

–equation (16 & 17): n_1 and n_2 are not defined

Response: We addressed this comment in the revised document (line 297).

–line 335: undefined reference

Comment has been addressed (line 308).